

# Thermodynamic and cloud evolution in a cold air outbreak during HALO-(AC)[3]: Quasi-Lagrangian observations compared to the ERA5 and CARRA reanalyses

Benjamin Kirbus[1], Imke Schirmacher[2], Marcus Klingebiel[1], Michael Schäfer[1], André Ehrlich[1], Nils Slättberg[3], Johannes Lucke[4], Manuel Moser[4], Hanno Müller[1], and Manfred Wendisch[1]

[1]Leipzig Institute for Meteorology, Leipzig University, Leipzig, Germany
[2]Institute of Geophysics and Meteorology, University of Cologne, Cologne, Germany
[3]Alfred Wegener Institute, Helmholtz Centre for Polar and Marine Research, Potsdam, Germany
[4]Institute of Atmospheric Physics, German Aerospace Center (DLR), Weßling, Germany

**Correspondence:** Benjamin Kirbus (benjamin.kirbus@uni-leipzig.de)

**Abstract.** Intense air mass transformations take place when cold, dry Arctic air masses move southward from the closed sea ice onto the much warmer ice-free Arctic ocean during marine cold air outbreaks (MCAOs). In spite of intensive research on MCAOs during recent years, the temporal rates of diabatic heating and moisture uptake relevant also for cloud formation/dissipation have not been measured along MCAO flows. Instead, reanalyses have typically been used for climatological

investigations of MCAOs or to supply higher resolution models with lateral boundary conditions and time-dependent forcings. Meanwhile, the uncertainties connected to those datasets remain unclear.

Here, we present height-resolved observations of diabatic heating rates, moisture uptake, and cloud evolution measured in a quasi-Lagrangian manner. The investigated specific MCAO was observed on 01 April 2022 during the HALO-(AC)[3] airborne campaign that was conducted in spring 2022. Shortly after passing the ice edge, maximum diabatic heating rates

larger than $6\,\mathrm{K\,h^{-1}}$ and moisture uptake of more than $0.3\,\mathrm{g\,kg^{-1}\,h^{-1}}$ were measured close above the ocean surface. As the air mass continued its drift southwards, clouds started to form and vertical mixing within the steadily deepening boundary layer was intensified. The quasi-Lagrange observations are compared with reanalysis data from the European Centre for Medium-Range Weather Forecasts (ECMWF) latest global reanalysis ERA5 and the Copernicus Arctic Regional Reanalysis (CARRA). It was found that the mean absolute errors (MAEs) of ERA5 versus CARRA data are $60\,\%$ higher for air temperature over

sea ice ($1.4\,\mathrm{K}$ versus $0.9\,\mathrm{K}$), and $70\,\%$ higher for specific humidity over ice-free ocean ($0.12\,\mathrm{g\,kg^{-1}}$ versus $0.07\,\mathrm{g\,kg^{-1}}$). We relate these differences not only to issues with representations of the marginal ice zone and corresponding surface fluxes in ERA5, but also to the cloud scheme producing excess liquid-bearing clouds and precipitation, causing a too-dry marine boundary layer. Overall, the combination of CARRA's high spatial resolution, an improved handling of cold surfaces, and the demonstrated higher fidelity towards the observations, make it a well-suited candidate for further investigations of Arctic air

mass transformations.





## 1 Introduction

Arctic air masses over closed sea ice are subject to a sustained radiative cooling. Therefore, they are characterized by cold temperatures and low atmospheric moisture contents. Marine cold air outbreaks (MCAOs) manifest when such Arctic air masses exit the closed sea ice, traverse the marginal sea-ice zone (MIZ) and ultimately move southward onto considerably warmer ice-free oceans (Fletcher et al., 2016a; Dahlke et al., 2022). During MCAOs, significant air mass transformations occur caused by strong surface energy fluxes of sensible and latent heat. This leads to intense diabatic heating, moisture uptake, a deepening of the atmospheric boundary layer (ABL), and roll convection that initiate cloud evolution (Fletcher et al., 2016a; Papritz and Spengler, 2017; Pithan et al., 2018). As a result, the near-surface air temperature can increase by more than 20 K in a matter of hours (Pithan et al., 2018; Wendisch et al., 2023a). Characteristic cloud streets of up to 1000 km length are formed, which later break up due to processes such as decoupling and precipitation formation (Fletcher et al., 2016a; Pithan et al., 2018; Lloyd et al., 2018; Tornow et al., 2021; Dahlke et al., 2022; Sanchez et al., 2022; Murray-Watson et al., 2023). In the temperature range of -25 °C to 0 °C, these clouds are often of mixed-phase type, where typically the upper portions of the clouds are dominated by supercooled liquid water and the lower parts by ice particles (Shupe et al., 2006; Morrison et al., 2012). The strongest MCAO events occur in winter, when the horizontal surface temperature gradient between the cold sea ice and the adjacent ice-free ocean is the largest (Fletcher et al., 2016a; Papritz and Spengler, 2017; Dahlke et al., 2022). One of the primary gateways into and out of the central Arctic is the Fram Strait, located between Greenland and the Svalbard archipelago. MCAOs are favored in this gateway because the North Atlantic Current transports significant heat northward, and consequently the MIZ and sea-ice edge are located far northward as well (Dahlke et al., 2022), which promotes intense MCAOs in this region (Papritz and Spengler, 2017).

Several factors have sparked scientific interest in studying MCAOs. The formation of cloud streets and their transition into open cells have important implications for the Arctic and the mid-latitude radiative energy budget, as the bright clouds over dark, ice-free ocean surfaces reflect a large fraction of incoming solar radiation, which causes a significant cooling at the surface (Li et al., 2011; Sanchez et al., 2022; Murray-Watson et al., 2023). Furthermore, large amounts of heat are transferred from the ocean into the atmosphere. Estimates show that about 60-80 % of oceanic heat loss in the Nordic Seas in winter is caused by MCAOs, which has important implications for deep water formation (Papritz and Spengler, 2017). MCAOs have been linked to the evolution of short-lived polar lows and mesoscale cyclones (Stoll et al., 2018; Landgren et al., 2019; Meyer et al., 2021; Terpstra et al., 2021). Either with or without such low-pressure systems being present, MCAOs can trigger extreme weather conditions, such as freezing sea spray, intense snowfall, or high near-surface winds. These phenomena pose significant hazards at affected coastlines (Kolstad, 2017; Landgren et al., 2019). The Arctic amplification observed in recent decades has caused significant reduction in strong wintertime MCAOs in the Fram Strait (Dahlke et al., 2022) and Barents Sea (Narizhnaya et al., 2020). Also in the future, strong wintertime MCAOs are expected to decrease (Landgren et al., 2019). On the contrary, springtime MCAOs are observed to intensify (Dahlke et al., 2022). Not only are the MCAO intensities expected to change, but the melting Arctic sea ice is also leading to a shift of spatial patterns (Landgren et al., 2019).



MCAOs have been studied intensively using satellite data (Sarkar et al., 2019; Christensen et al., 2020; Wu and Ovchinnikov, 2022b; Murray-Watson et al., 2023; Mateling et al., 2023b), atmospheric soundings (Dahlke et al., 2022; Geerts et al., 2022; Michaelis et al., 2022), and dedicated (mostly airborne) field campaigns (such as reported by Brümmer, 1996; Geerts et al., 2022; Sanchez et al., 2022; Mech et al., 2022a; Michaelis et al., 2022; Sorooshian et al., 2023). The models applied to represent MCAOs range from turbulence-resolving large eddy simulations (Tomassini et al., 2017; Tornow et al., 2021; Li et al., 2022), mesoscale numerical weather prediction models (Tomassini et al., 2017; Field et al., 2017), to global climate models (Kolstad and Bracegirdle, 2007; Smith and Sheridan, 2021).

In addition, sophisticated atmospheric reanalyses have been developed. They assimilate a large amount of available measurements, such as atmospheric soundings and satellite data (Hersbach et al., 2020). Reanalyses deliver meteorological parameters on a continuous latitude/longitude/height grid, as well as on high temporal resolution down to 1 hour. The fifth generation atmospheric reanalysis (ERA5) of the European Centre for Medium-Range Weather Forecasts (ECMWF) is frequently used for climatological studies (Papritz and Spengler, 2017; Papritz et al., 2019; Dahlke et al., 2022). Furthermore, dedicated Arctic reanalyses have been developed, such as the spatially much higher resolved Copernicus Arctic Regional Reanalysis (CARRA). Investigations into characteristic properties and trends of Arctic MCAOs based on reanalyses have been created for classical Eulerian (Dahlke et al., 2022) and quasi-Lagrangian frameworks (Papritz and Spengler, 2017). 'Quasi'-Lagrangian highlights the fact that an air mass is not truly physically followed, as it may be possible by meteorological balloons (Businger et al., 2006). Instead, wind fields as available from reanalyses are used to model the flow of air masses (Sprenger and Wernli, 2015), and then for example aircraft are employed to trace specific air mass parcels along their trajectory (Boettcher et al., 2021; Sanchez et al., 2022). In addition, reanalyses are used to supply the boundary conditions and time-dependent forcings to much higher resolved models (Seethala et al., 2021; Li et al., 2022).

However, microphysical properties and the processes governing the evolving cloud (radiative) properties remain notoriously difficult to model (Pithan and Mauritsen, 2014; Pithan et al., 2018; Wendisch et al., 2021). This is especially true over sea ice and the MIZ, where the widely employed satellite-based remote sensing faces serious challenges. As a result, many satellite studies investigating MCAOs focus solely on the evolution over the fully ice-free open ocean (Wu and Ovchinnikov, 2022a; Murray-Watson et al., 2023; Mateling et al., 2023b). Furthermore, the vertically non-uniform diabatic heating and moisture uptake by air masses along MCAO trajectories are not sufficiently represented in models, which may cause issues in terms of atmospheric stability and the lapse-rate feedback (Linke et al., 2023). While the contributing processes are generally well understood, their relative importance and absolute magnitudes remain unspecified (Pithan et al., 2018; Wendisch et al., 2021; You et al., 2021a, b). As a result, the overall cloud effects on Arctic climate remain uncertain (Boucher et al., 2014; Wendisch et al., 2021, 2023b).

Here, we present airborne measurements of the height-dependent heating and moistening rates during a specific MCAO event, based on quasi-Lagrangian airborne observations. The investigated flight of the High Altitude and LOng Range Aircraft (HALO) was conducted as part of the HALO-(AC)[3] airborne campaign, which took place in spring 2022. We compare the quasi-Lagrangian observations to the ERA5 and CARRA reanalyses. In our article, we address three specific research questions: (Q1) How do air temperature, specific humidity, and clouds evolve in the first four hours of the developing MCAO? (Q2)





How do the ERA5 and CARRA reanalyses perform with respect to observations, and compared to each other? (Q3) What are
possible sources of errors, which could explain deviations between reanalysis output and observations?

The study is structured as follows. Section 2 details the airborne observations, which form the basis of this study. The two
ERA5 and CARRA reanalyses are introduced, and the trajectory analysis is described. In Section 3, the airborne measurements
are analyzed in a classical Eulerian framework. Subsequently, the quasi-Lagrangian analysis will be used to present and dis-
cuss novel observation-derived heating and moistening rates along the MCAO flow, correlated cloud properties, as well as to
compare them between the two reanalyses.

## 2 Methods

### 2.1 Airborne observations

The MCAO analyzed in this study was observed on 01 April 2022 during the HALO-(AC)[3] campaign, which was conducted
in March and April 2022 as dedicated quasi-Lagrangian Arctic airborne campaign (Wendisch et al., 2021, 2023c). The me-
teorological conditions that prevailed during the campaign are described in Walbröl et al. (2023). HALO-(AC)[3] involved the
German Aerospace Center's HALO (Stevens et al., 2019; Ehrlich et al., 2023) for the long-range investigation of air mass
transformations in combination with the lower-flying Polar 5 and Polar 6 research aircraft operated by the Alfred Wegener
Institute, Helmholtz Centre for Polar and Marine Research (Wesche et al., 2016). After taking off from the base in Kiruna
(Sweden) at 07:30 UTC, HALO headed north. It then sampled the MCAO cloud streets west of Svalbard, see Figure 1. The
speed of HALO at its typical flight altitude of 10-12 km is around 800 km h$^{-1}$, which is much faster than the wind speed of
30-60 km h$^{-1}$ measured by dropsondes on this day. Therefore, in order to facilitate a quasi-Lagrangian (i.e., air mass following)
sampling of cloudy air masses, long horizontal cross-sections were flown across the off-ice flow. These flight legs not only cov-
ered the ice-free ocean, but also parts of the adjacent Arctic sea ice, see Figure 1. Several of such flight legs were conducted,
where the legs were step-wise shifted south roughly according to the forecast wind speed in the atmospheric boundary layer.
Similar quasi-Lagrangian airborne sampling was reported previously, however taking place over the Atlantic (Methven et al.,
2006) and for a warm conveyor belt over Europe (Boettcher et al., 2021). Similar to our case, Sanchez et al. (2022) investigated
the aerosol and cloud evolution in MCAOs. From their quasi-Lagrangian observations, they contrast the evolving particle mode
distributions between within and outside MCAO flow. However, they do not report e.g. on heating or moistening rates.





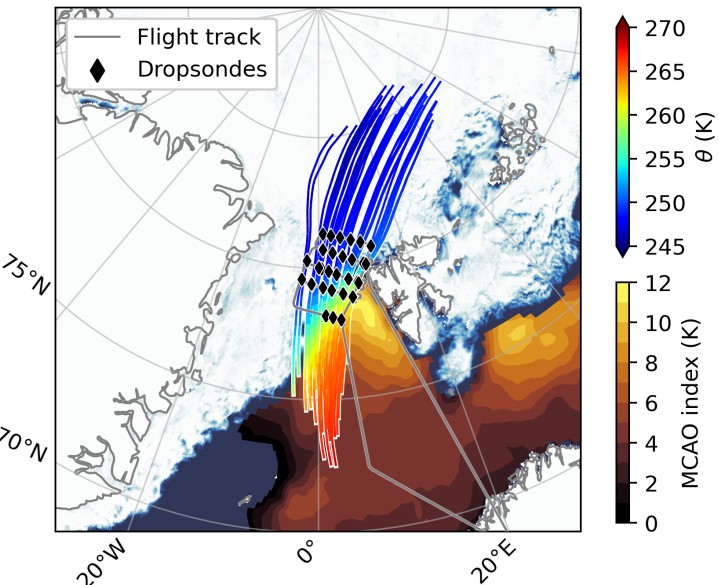

**Figure 1.** Case overview. The gray line shows the flight track of HALO on 01 April 2022. Diamond shapes show the locations where dropsondes were released. White-blueish contours represent 1 km high-resolution sea-ice concentration retrieved from a merged MODIS-AMSR2 satellite product (Ludwig et al., 2020). Over the ice-free ocean, yellow-brownish contours indicate the ERA5-derived MCAO index $M_{850\,\mathrm{hPa}}$, see Equation 3. Finally, the colored lines show 24 hours backwards and 24 hours forward trajectories initialized at the location of each dropsonde at 10 hPa above ground. The colors represent the evolving potential temperature ($\theta$) of these air masses as traced from ERA5 data.

Northwest of Svalbard, a total of 40 dropsondes were released from HALO. These RD94 dropsondes recorded air pressure

$p$ (accuracy $0.4\,\mathrm{hPa}$), air temperature $T$ ($0.2\,\mathrm{K}$), relative humidity $RH$ ($2\,\%$), derived potential temperature $\theta$ and specific humidity $q$, as well as horizontal wind components (Vaisala, 2010; George et al., 2021). The data were assimilated by the ECMWF Integrated Forecasting System (IFS), also serving as input for the ERA5 and CARRA reanalyses. The profiles of $\theta$ are used to derive the atmospheric boundary layer height (BLH) from dropsonde measurements and reanalyses. The BLH is defined here as altitude where the largest gradient in $\theta$ is found (similar as in Seidel et al., 2010; Dai et al., 2014; Bakas et al.,

2020; Sinclair et al., 2022). For estimating the cloud top heights (CTHs), the 532 nm back-scatter ratio from the Water Vapour Differential Absorption (WALES) lidar is used (Wirth et al., 2009; Ehrlich et al., 2023). WALES has a vertical resolution of 15 m. We define the CTH as the maximum altitude above ground where the back-scatter ratio exceeds that of cloud-free sections. To better understand the heating and moistening rates, airborne observations from HALO are used to estimate the surface sensible and latent heat fluxes (SSHF, SLHF), similar to Li et al. (2022). SSHF and SLHF are calculated as follows

(ECMWF, 2016):

$$\mathrm{SSHF} = \rho_{\mathrm{air}}\, C_{\mathrm{H}}\, \mathrm{c_p}\, |U_{10\mathrm{m}}|\, (T_{10\mathrm{m}} - T_{\mathrm{skin}}), \tag{1}$$



$$\text{SLHF} = \rho_{\text{air}} \, C_{\text{Q}} \, L_{\text{v}} \, |U_{\text{10m}}| \, (q_{\text{10m}} - 0.98 \, q_{\text{sat,skin}}) , \tag{2}$$

where $\rho_{\text{air}}$ denotes the air density (kg m$^{-3}$), $C_{\text{H}}$ and $C_{\text{Q}}$ are the transfer coefficients for heat and humidity (dimensionless), $c_{\text{p}}$ is the specific heat capacity at constant pressure ($c_{\text{p}} = 1004.7 \, \text{J kg}^{-1} \, \text{K}^{-1}$), $L_{\text{v}}$ the latent heat of evaporation ($L_{\text{v}} = 2.5008 \, \text{J kg}^{-1}$), $U_{\text{10m}}$ the wind speed at 10 m height (m s$^{-1}$), $(T_{\text{10m}} - T_{\text{skin}})$ the temperature difference between the 10 m air temperature and skin temperature (K), and $(q_{\text{10m}} - 0.98 \, q_{\text{sat,skin}})$ the difference in specific humidity between the 10 m level and the specific saturation humidity taken at skin temperature ( kg kg$^{-1}$). The factor 0.98 accounts for the reduction in vapor pressure resulting from a typical sea-water salinity of 3.4 % (Fairall et al., 2003; ECMWF, 2016). Dropsonde profiles are used to extract $\rho_{\text{air}}$, $U_{\text{10m}}$, $T_{\text{10m}}$, and $q_{\text{10m}}$ via linear interpolation to the 10 m height level. The Video airbornE Longwave Observations within siX channels (VELOX) thermal infrared imager (Schäfer et al., 2022) is applied to obtain $T_{\text{skin}}$ and $q_{\text{sat,skin}}$ for the cloud-free sections. The transfer coefficients of heat and humidity are taken from ERA5.

To collect in situ cloud measurements, the Polar 6 aircraft sampled concurrently with HALO (Figure A2 in the Supplement). Polar 6 was based in Longyearbyen on Svalbard and was equipped with a wide range of in situ probes (Moser et al., 2023), including a Nevzorov sonde from which the liquid and frozen cloud water contents were obtained (Korolev et al., 1998; Lucke et al., 2022; Mech et al., 2022a).

## 2.2 Reanalysis products

The ERA5 global reanalysis features a sophisticated four-dimensional variational data assimilation scheme (Hersbach et al., 2020) and is based on ECMWF's IFS Cycle 41r2 (ECMWF, 2016). ERA5 data fields have a temporal resolution of one hour, a horizontal grid resolution of 31 km, and a vertical resolution of 137 model levels. Several studies note the high performance of ERA5 in the Arctic region (Graham et al., 2019a; Wu et al., 2023), specifically in the Fram Strait region (Graham et al., 2019b). Thus, numerous authors performing trajectory analysis in the Arctic rely on wind and meteorological data fields from ERA5 (e.g., Papritz and Spengler, 2017; Papritz, 2020; Dahlke et al., 2022; You et al., 2021b; Kirbus et al., 2023a, b; Svensson et al., 2023). In this study, ERA5 wind fields are used for trajectory calculations, thermodynamic profiles are extracted at the dropsonde locations, and several cloud-related parameters and turbulent energy fluxes are retrieved.

The CARRA regional reanalysis was specifically tailored towards the unique conditions in the Arctic environment, such as the prevailing cold surfaces on Arctic sea ice and ice sheets. Notably, it explicitly simulates a snow layer on sea ice. CARRA is based on the HARMONIE-AROME non-hydrostatic regional numerical weather prediction model, which is operational in the Nordic and several other European countries (Bengtsson et al., 2017; Yang et al., 2023). It can be retrieved for two distinct domains (CARRA-West covering Greenland, CARRA-East encompassing Svalbard and Northern Scandinavia) that overlap in the vicinity of Svalbard (Yang et al., 2023). Boundary forcings are taken from ERA5. CARRA analysis fields have a temporal resolution of three hours, a horizontal grid resolution of 2.5 km, and 65 vertical model levels.

Compared to ERA5, a larger amount of local observations is input into CARRA's three-dimensional variational assimilation scheme, such as snow depths from satellite observations or actual measurements of glacier albedos. Satellite-borne sea-surface





temperature and sea ice data are assimilated at a higher spatial resolution compared to ERA5. Especially in areas with steep topography, the increased resolution of CARRA versus ERA5 is expected to better fit to observations (Yang et al., 2023). Isaksen et al. (2022) show that both reanalyses reproduce the key features of the observed exceptional warming over the Barents Sea. However, CARRA shows more spatial details and larger regional surface air temperature trends. Moore and Imrit (2022) investigate winds in the 40-100 km narrow Nares Strait northwest of Greenland. They find a significant underestimation

of local wind speeds in ERA5, which on average reach 40 % of the observed values versus 80 % in CARRA. Box et al. (2023) evaluate five contemporary numerical prediction systems against in situ rainfall data from Greenland stations. CARRA shows the lowest average bias and the highest explained variance. Køltzow et al. (2022) systematically check the representation of 10-m wind speed and 2-m air temperature against observations for the two CARRA domains. The largest differences between CARRA and ERA5 are found in regions with complex terrain and coastlines, as well as over the Arctic sea ice for 2-m air

temperature in winter. Over flat terrain, the added value is especially obvious for the air temperature. With these reported advantages in mind, CARRA focuses solely on the European Arctic sector and starts only in 1991. The three-hourly analysis fields must be combined with short-range forecasts to match the same one-hourly resolution of ERA5 (Yang et al., 2023).

To classify the strength of the observed MCAO, the Marine Cold Air Outbreak index $M$ (Kolstad et al., 2009; Fletcher et al., 2016b) is calculated based on ERA5 data and a 850 hPa reference level (Papritz et al., 2015; Papritz and Spengler, 2017; Knudsen et al., 2018; Dahlke et al., 2022; Geerts et al., 2022; Mateling et al., 2023a). Using the potential temperature $\theta$,

$M_{850\,\mathrm{hPa}}$ is computed as follows:

$$M_{850\,\mathrm{hPa}} = \theta_{\mathrm{skin,ocean}} - \theta_{850\,\mathrm{hPa}}, \tag{3}$$

where $\theta_{\mathrm{skin,ocean}}$ denotes the potential skin temperature over ice-free ocean. A positive $M_{850\,\mathrm{hPa}}$ over a large area hints toward the presence of a MCAO event. The daily $M_{850\,\mathrm{hPa}}$ is averaged temporally from the hourly input data, and spatially

over a box surrounding Fram Strait. With an extent of 75-80 °N and 10 °W-10 °E, this box is consistent with previous studies (Papritz and Spengler, 2017; Dahlke et al., 2022). Consistent with the aforementioned works, MCAO events can be classified as weak ($M_{850\,\mathrm{hPa}}$ below 4 K), moderate ($M_{850\,\mathrm{hPa}}$ between 4-8 K), or strong ($M_{850\,\mathrm{hPa}}$ above 8 K).

### 2.3 Trajectory analysis

To check whether the quasi-Lagrangian flight strategy on 01 April 2022 had been a success, the ERA5 three-dimensional wind

fields are retrieved on 137 model levels. Note that all 40 released dropsondes were assimilated by ECMWF, which greatly improves the reliability of trajectory calculations. ERA5 analysis wind fields are chosen over CARRA's as they exhibit a slightly higher temporal and vertical resolution, but no significant differences are found when using CARRA data (a comparison of wind profiles is given in Figures A3 and A4). Similarly, Køltzow et al. (2022) also reported only small differences between ERA5 and CARRA wind fields in areas with flat terrain, such as over the Arctic Ocean. *Lagranto* (Sprenger and Wernli,

2015) is then used to identify quasi-Lagrangian matches, where the same air masses were sampled within a 20 km radius below HALO twice, first at times $t_1$ and then again at $t_2$. Air masses are initialized every 1 min along HALO's flight track, vertically every 5 hPa between 250 hPa and the surface, and horizontally evenly spaced every 7 km in a 20 km radius. A match



is registered if the same air mass is seen again in the column below HALO within the same 20 km radius. More details are found in the appendix and in Wendisch et al. (2023c). As matches are altitude-dependent, from the closest dropsonde the nearest potential air temperature and specific humidity measurements are retained. Potential temperature is chosen instead of regular air temperature to focus on diabatic processes (Papritz and Spengler, 2017; Dahlke et al., 2022). Applying all filters yields approx. 24,200 quasi-Lagrangian matches. The net diabatic heating and moistening rates are calculated as:

$$\left(\frac{\Delta\theta}{\Delta t}\right)_{\mathrm{net}} = \frac{\theta_2 - \theta_1}{t_2 - t_1}, \tag{4}$$

$$\left(\frac{\Delta q}{\Delta t}\right)_{\mathrm{net}} = \frac{q_2 - q_1}{t_2 - t_1}. \tag{5}$$

The air mass transformations occurring in MCAOs are primarily forced by the transition from closed sea ice to ice-free ocean (Pithan et al., 2018; Wendisch et al., 2023a). Therefore, the quasi-Lagrangian matches are grouped by the time each air mass has spent over ice-free ocean. For all dropsonde locations, 12 hour backward trajectories are calculated for the air masses in the lowest 10 hPa (approx. 100 m) above ground. The sea-ice concentration (SIC) is traced along each trajectory (Figure A1). For this purpose, the merged MODIS and ASI-AMSR2 data at 1 km grid resolution generated by the University of Bremen (Ludwig et al., 2020) is interpolated to a 0.05°x0.05° latitude/longitude grid. The duration over ocean is defined as the time the air mass spent over ice-free ocean (sea-ice concentration SIC≤20 %) until it first reaches a SIC>20 %.

While the flight leg of Polar 6 on 01 April 2022 was aligned in parallel with HALO's center leg, it still covered different regions at different times than HALO, not least due to the much lower speed of Polar 6 of around 300 km h[-1]. To make data comparable, the same approach is taken as for HALO: Every 1 min along the flight track, air masses are initialized. However, due to the in situ sampling method, the air masses are started at the actual flight level of Polar 6 and SIC traced (Figure A2). As a result, the in situ observations are transformed into the same coordinate system of "time over ocean" as for HALO, which is the assumed primary driver of the observed air mass transformations. Due to its limited range, Polar 6 only sampled the first three hours of the MCAO.

## 3 Results and discussion

### 3.1 Case overview

Figure 1 gives an overview of the conditions on 01 April 2022. Depicted is the flight track of HALO as well as the dense grid of dropsondes released west of Svalbard. The daily averaged MCAO index $M_{850\,\mathrm{hPa}}$ in the Fram Strait box is found to be 7.7 K. This qualifies the MCAO investigated here between a moderate and strong case, following the classification of Papritz and Spengler (2017) and Dahlke et al. (2022). According to the ERA5-based MCAO climatology 1979-2020 by Dahlke et al. (2022), the median daily frequency of occurrence for MCAOs in Fram Strait is at around 50-70 % both in March and April. Furthermore, events of similar magnitude can be expected at around 40 % of all days (Dahlke et al., 2022). This means that on 01 April 2022, HALO sampled a quite typical event for this region and time of the year. Figure 1 also reveals a maximum





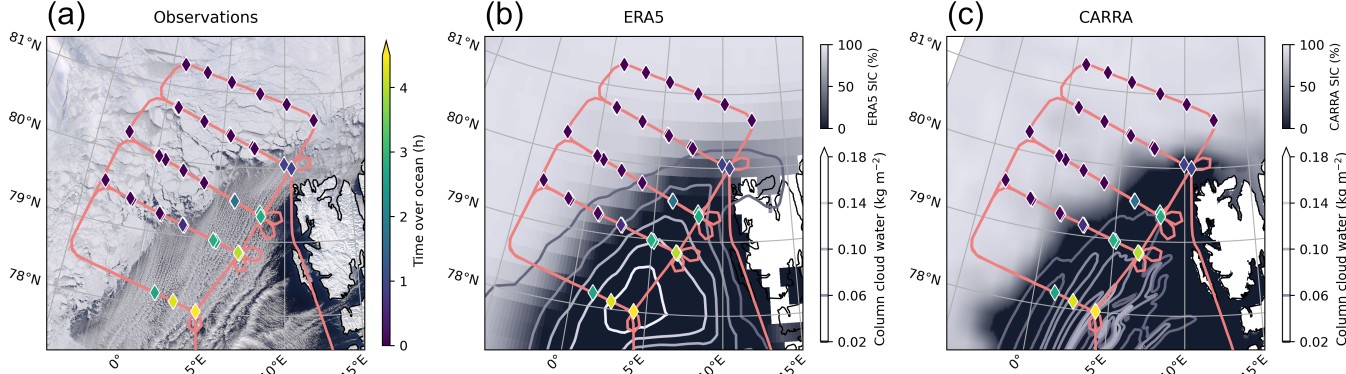

**Figure 2.** Sea ice and cloud structures to the northwest of Svalbard on 01 April 2022 based on observations and reanalyses. In each subplot, the red line shows the flight path of HALO, and diamond shapes show the location of released dropsondes. The shapes are colored by time near-surface air masses spent over ice-free ocean (Sea-ice concentration SIC below 20 %; Ludwig et al. (2020)). a) The Terra/MODIS corrected reflectance shows the formation of cloud streets shortly after the off-ice drift. The image is taken from NASA Worldview (2023). b) ERA5 data at 12 UTC. Shown is the SIC (filled contours) and the total column cloud liquid + ice water (contour lines). c) CARRA data at 12 UTC. Depicted are SIC (filled contours) and the total column cloud liquid + ice + graupel water (contour lines).

$M_{850\,\mathrm{hPa}}$ of above 12 K close to the marginal sea-ice zone. This highlights the strong temperature contrasts that the cool Arctic
air masses experience when departing the closed Arctic sea ice.

To better comprehend the air mass flow, a set of 40 trajectories is initialized at the location of each dropsonde with 1 min temporal resolution. These trajectories are started at 10 hPa above ground and calculated both forwards and backwards in time over a 24-hour period. The ERA5-derived potential temperature is then traced. During their drift over closed Arctic sea ice, the near-surface air parcels do not undergo any significant diabatic temperature changes. However, once they cross the MIZ and
reach the ice-free ocean, the air masses undergo a rapid diabatic heating of up to 20 K within 24 hours.

## 3.2 Eulerian comparison of observations and reanalyses

### 3.2.1 Sea ice and cloud structures

Figure 2 depicts a first comparison between observations and the reanalyses. Figure 2a shows the Terra/MODIS corrected reflectance from NASA Worldview for 01 April 2022 (NASA Worldview, 2023). From the satellite imagery, it becomes clear
that the Arctic sea ice northwest of Svalbard features many leads on different length scales. However, the MIZ is rather sharp, and the transition from closed sea ice to ice-free ocean water typically occurs within less than 1 km distance. Over the ice-free ocean, cloud streets due to roll convection are obvious. The cloud streets form along to the prevailing wind direction. Furthermore, a clear Lee effect due to Svalbard's mountain ranges is seen to the west of the archipelago.

Figure 2b shows the corresponding fields as represented by ERA5 at 12 UTC noon. Due to its coarse spatial resolution, no
leads are modeled in the SIC data fields, and the MIZ width is on a length scale of approximately 80 km. This is a typical MIZ



width for ERA5 (Renfrew et al., 2021). Instead of cloud streets, a stratiform liquid+ice containing cloud deck is simulated, which thickens in off-ice direction. Clouds are partly also already formed over closed sea ice. In contrast, clouds in CARRA are exclusively formed over the ice-free ocean, see Figure 2c. The high spatial resolution allows convection to be modeled. As a result, several distinct cloud streets are reproduced. In addition, CARRA better reproduces the sharp MIZ, which here is

on the scale of around 10 km. The sharper MIZ of CARRA in comparison to ERA5 is not only an issue of spatial resolution (2.5 km for CARRA versus 30 km for ERA5). The sea-ice concentrations of ERA5 is derived from the Operational Sea Surface Temperature and Ice Analysis dataset, produced by the UK Met Office (OSTIA; Donlon et al., 2012). OSTIA outputs daily sea-surface temperature and sea-ice concentration fields based on satellite observations, with a native resolution of 0.05°x0.05° (roughly 6 km). Yet the sea-ice data is based on the EUMETSAT OSI-SAF 401 dataset utilizing 19 GHz and 37 GHz microwave

channels at along-track resolutions of coarse 69 and 37 km (Tonboe et al., 2017; Renfrew et al., 2021). Several authors noted that improved sea-ice and MIZ representation crucially improve the performance of models in the lower-tropospheric layers (Liu et al., 2006; Gryschka et al., 2008; Chechin et al., 2013; Müller et al., 2017; Spensberger and Spengler, 2021). As will be shown later, the magnitude of turbulent heat fluxes is directly correlated to the distribution of sea-ice versus ice-free ocean, which is the primary driver of MCAO transformations. Therefore, errors e.g. in MIZ width can have significant downstream

effects over several hundreds of kilometers (Tomassini et al., 2017; Spensberger and Spengler, 2021).

### 3.2.2  Vertical thermodynamic profiles

Figure 3a shows the profiles of air temperature from observations. Over sea ice, clear temperature inversions are found. The coldest near-surface temperatures reach -27 °C, and the thickness of the inversions is around 0.6-0.9 km. As air masses spend more time over ice-free waters, they become warmer near the surface, leading to stronger coupled ABLs and the development

of a typical marine stratification. This is accompanied by a steady increase in the calculated BLHs and closely correlated CTHs.



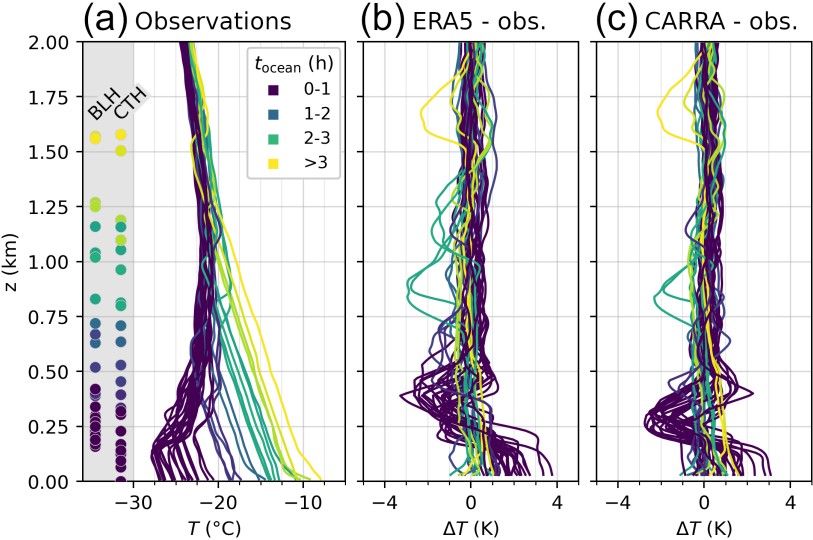

**Figure 3.** Vertical profiles of air temperature ($T$) in the lowest 2 km above ground taken from observations and reanalyses. In all panels, profiles are colored by the time air masses spent over open ocean. a) Observed profiles of air temperature. Measurement-derived atmospheric boundary-layer heights (BLHs) and cloud top heights (CTHs) are indicated to the left hand side. b) Deviation of the ERA5 profiles from the observed profiles, and c) Deviation of the CARRA profiles from the observed profiles.

By linearly interpolating all data to 100 m vertical resolution, the temperature differences $\Delta T$ between ERA5/CARRA and the observations is computed. The results are shown in Figures 3b and 3c. Despite the reanalysis assimilating all the employed dropsondes, the ERA5 profiles show a distinct warm bias in near-surface air temperatures of mean 2 K over Arctic sea ice. Many authors reported on similar warm biases of skin and near-surface air temperatures in ERA5 (Batrak and Müller, 2019;
Wang et al., 2019; Tjernström et al., 2021; McCusker et al., 2023). The skin temperatures are generally considered too warm as an insulating layer of snow is missing atop the floating ice, which can introduce surplus heat into the lower atmosphere (Batrak and Müller, 2019; Wang et al., 2019). The surface warm bias turns to a mean cold bias of -1 K at altitudes of 0.25-0.50 km. Over the ocean, the mean temperature bias is much lower and reaches -0.5 K at around 1 km altitude. In the CARRA data, the near-surface temperature bias is reduced to an average of 1 K. Similar improvements over ERA5 have been reported by
others (Køltzow et al., 2022). However, CARRA also faces challenges in accurately representing temperature inversions. This is reflected in the nearly identical cold bias of -1 K, yet at slightly lower altitudes of 0.20-0.40 km.

The mean absolute errors (MAEs) of ERA5 and CARRA with regard to measurements are computed. To evaluate especially the crucial ABL representation, MAEs are averaged vertically from surface up to the respective observations-derived BLH. Table 1 summarizes the results separately for dropsondes released over sea ice and ice-free ocean. For air temperature over ice,
CARRA clearly shows a smaller MAE of 0.9 K versus 1.4 K for ERA5. Over the ice-free waters of Fram Strait, these errors are significantly reduced in both products, yielding a comparable MAE of around 0.3 K.





**Table 1.** Mean absolute errors (MAEs) of ERA5 and CARRA profiles compared to observations. The MAEs are averaged vertically up to the observed boundary-layer heights. Results are shown for the variables air temperature ($T$) and specific humidity ($q$), grouped by surface type. Profiles are classed as 'sea ice' ('open ocean') if the AMSR2 sea-ice concentrations is above (below) 50 %.

| Variable | Surface | MAE of ERA5 | MAE of CARRA |
|---|---|---|---|
| $T$ | sea ice | 1.4 K | 0.9 K |
| | open ocean | 0.3 K | 0.3 K |
| $q$ | sea ice | 0.03 g kg$^{-1}$ | 0.03 g kg$^{-1}$ |
| | open ocean | 0.12 g kg$^{-1}$ | 0.07 g kg$^{-1}$ |

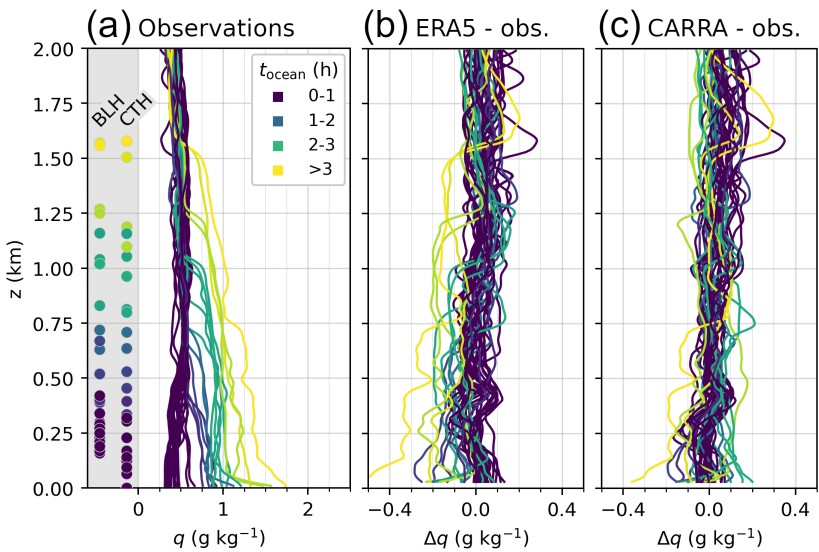

**Figure 4.** Same as Figure 3, but for specific humidity in the lowest 2 km above ground. a) Observed profiles of specific humidity. b) Deviation of the ERA5 profiles from the observed profiles, and c) Deviation of the CARRA profiles from the observed profiles.

Next, the vertical profiles of specific humidity are examined. Figure 4a depicts the observed profiles, as extracted from the dropsonde measurements. Over sea ice, a uniform and dry ABL is found, where maximum values of around 0.6 g kg$^{-1}$ are measured. Near-surface layers are the driest, at around 0.4 g kg$^{-1}$. The longer the air masses reside over the sea, the more water vapor is picked up by the lower air mass layers through evaporation from the ocean surface.

Over sea ice, ERA5 shows a mean near-surface moist bias of 0.05 g kg$^{-1}$ (Figure 4b), as well as a slight dry bias close to the BLHs. Once air masses drift over the sea, a strong dry bias is found throughout the ABL. It increases over time and reaches down to -0.5 g kg$^{-1}$, which corresponds to about 30 % of the observed values. CARRA shows different patterns (Figure 4c). Over the closed ice pack, the lowest 0.2 km show a negligible humidity bias. However, in higher layers above 0.5 km, a slight moist bias is seen. During the off-ice drift, at first a slight moist and later dry bias becomes obvious; however, this is much smaller compared to the ERA5 reanalysis. The same patterns are found in the quantified MAEs within the ABLs, see again



Table 1. Here, CARRA always performs better than ERA5. This is seen much more pronounced over ocean, where CARRA's MAE of around 0.07 g kg$^{-1}$ is much lower than ERA5's 0.12 g kg$^{-1}$.

### 3.3 Quasi-Lagrangian comparison of observations and reanalyses

#### 3.3.1 Quasi-Lagrangian matches

Figure 5 gives an overview of the quasi-Lagrangian matches calculated with reference to the dropsondes. All matches are colored by the time air masses spent over ice-free ocean. As described in the Methods (Section 2), these approximately 24,200 matches are a function of height above ground because not only are the zonal and meridional winds height-dependent, but also the vertical velocity is used on the three-dimensional trajectory calculations. This allows air masses to ascend or descend along 295 their horizontal flow. The matches cover 150 km along the prevailing wind direction over the Arctic sea ice, and about 200 km along the MCAO evolution over ice-free ocean.

Naturally, the question arises on how reliable the trajectory calculations presented here are. In previous studies, sometimes additional criteria were applied to prove the reliability of trajectories. These are similar hydrocarbon fingerprints between matches (Methven et al., 2006) or an inert perfluoromethylcyclopentane tracer being deployed (Boettcher et al., 2021). How- 300 ever, such a use of tracers is only possible in case of in situ sampling. In the MCAO presented here, the assimilation of the high-density grid of dropsondes serves as crucial input for ERA5 and CARRA. This can also be seen in the comparison of wind profiles as shown in Figures A3 and A4. With the exception of the nearest-surface layers, a close match is seen between dropsondes and both reanalyses. Also, as trajectories are only calculated over short spatiotemporal scales (on the order of 1-4 hours, 50-200 km), small errors can not add up as much. For some research questions, it might be more valuable to inves- 305 tigate transformations over larger spatio-temporal scales, such as was done for example for aerosol and hydrocarbon species (Methven et al., 2006; Sanchez et al., 2022). However, as will be demonstrated, the highly important thermodynamic evolution occurs on time scales of a few hours and below. If too much time passes between two matching observations, the 'net' rates, e.g. of $(\Delta\theta/\Delta t)_{\text{net}}$ and $(\Delta q/\Delta t)_{\text{net}}$ introduced in Equations 4 and 5, would smooth out short-lived effects even more; models would be increasingly needed to disentangle the net rates calculated over longer time frames into sections of more or less 310 intense transformations. Finally, the stochastic approach presented here of initializing and then registering matches for a large number of trajectories within a radius of 20 km around HALO's location is also essential to weigh matches by their frequency of occurrence. Notably, all deviations seen between the aforementioned observed and modeled wind profiles result in an error of less than this 20 km radius over 1-3 hours of drift.



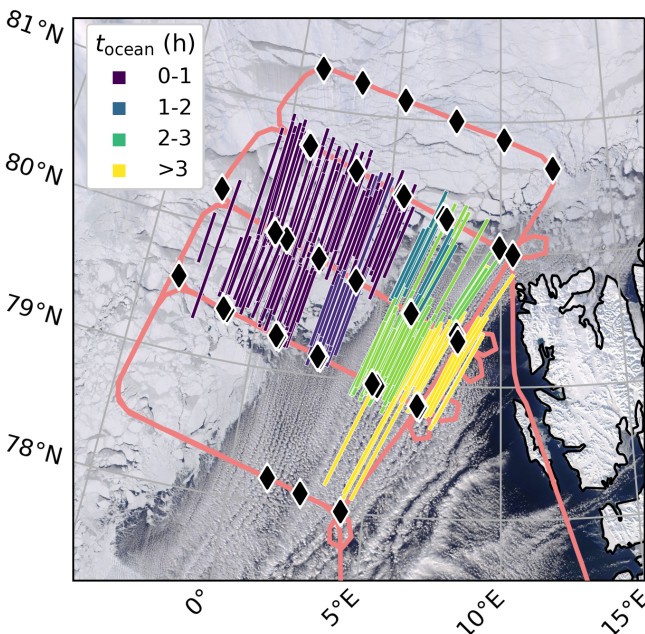

**Figure 5.** Spatial overview of the location of matching trajectories, which were calculated with respect to the dropsondes. Matching lines are colored by the time air masses spent over ocean. The background Terra/MODIS satellite image is taken from NASA Worldview (2023).

### 3.3.2 Diabatic heating and moistening rates

The evolution of thermodynamic properties in the form of heating and moistening rates is analyzed in a quasi-Lagrangian framework, grouped by time air masses spent over ice-free ocean.



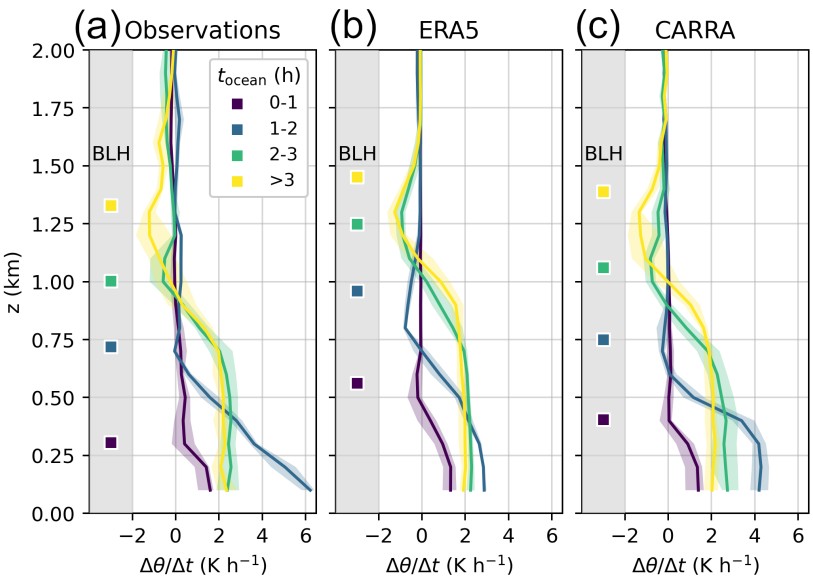

**Figure 6.** Diabatic heating rates grouped by time air masses spent over the ice-free ocean. At the left side of each panel, the mean BLH in each class is plotted as square. Lines depict the mean values, shading the 25-75 quantiles. a) Heating rates based on the quasi-Lagrangian dropsonde observations. b) Corresponding heating rates extracted from ERA5. c) Corresponding heating rates extracted from CARRA.

First, the evolution of diabatic heating rates is analyzed. Figure 6a shows the vertically resolved diabatic heating rates based on the quasi-Lagrangian dropsonde observations. Note that air parcels of the first category ($t_{ocean} = 0$-1 h) are drifting almost exclusively over sea ice and leads (Figure 5). For these air masses, a maximum near-surface warming of around $1.8\,\mathrm{K\,h^{-1}}$ is found. This possibly stems from some of the leads crossed by the trajectories. However, the heat is contained within the very shallow ABL, and all heating rates above the BLHs of around $0.30\,\mathrm{km}$ are around $0\,\mathrm{K\,h^{-1}}$. After crossing the MIZ and reaching the ice-free ocean ($t_{ocean} = 1$-2 h), this picture changes dramatically. A very intense surface warming is seen, where values larger than $6\,\mathrm{K\,h^{-1}}$ are found. This heating is starting to be mixed upwards into the increasingly deep ABL, which reaches BLHs of around $0.70\,\mathrm{km}$. After the initial rapid exposure of the cold and dry Arctic air masses to the much warmer ocean surface, the heating in the lowest layers declines rapidly and stays around $2\,\mathrm{K\,h^{-1}}$. The vertical mixing is now dominating and leads to almost homogeneous mixing within the lowest $0.75\,\mathrm{km}$ of the troposphere. Interestingly, some layers above show regions with negative heating rates, i.e., a net cooling of air masses at altitudes around the BLHs. An analysis of ERA5 temperature tendencies indicates that this is not a sign of a net cloud-top radiative cooling effect, but instead of the upward mixing of colder air into the original, overlying warmer inversion (Figure A5).

The corresponding ERA5-derived heating rates reflect the general features of the observations (Figure 6b). This is expected, as the 40 dropsondes were assimilated into ERA5. However, some important differences are found. Over the Arctic sea ice, ERA5 shows BLHs almost twice as high as seen in observations. While the slight surface warming of around $1.6$-$1.8\,\mathrm{K\,h^{-1}}$ is also seen in ERA5, it shows excess heat that it mixes upwards towards the BLH. The intense surface warming at $t_{ocean} = 1$-2 h





is not represented in ERA5. This is not surprising, as already the overview map in Figure 2b revealed a wide MIZ on the order

of 80 km, much wider than shown in the observations. As a result, the initial stage of the MCAO is delayed in ERA5. The later

stages ($t_{ocean} > 2$ h) of the MCAO, however, are rather well represented, yet again with an exaggerated vertical mixing. The

essential feature of negative heating rates in higher altitudes is captured.

Figure 6c shows the heating rates extracted from the CARRA product. Generally, these settle in between the observations

and ERA5. All BLHs are lower than in ERA5, and significantly closer to the observed values. For $t_{ocean} = 1$-2 h, the observed

intense warming rate larger than 6 K h$^{-1}$ is also not represented fully, yet much better than in ERA5. A maximum value for the

near-surface heating of around 4 K h$^{-1}$ is found, which is homogeneously mixed upwards up to 0.40 km altitude. This might in

part be caused by the much sharper MIZ, which is closer to reality (Figure 2c).

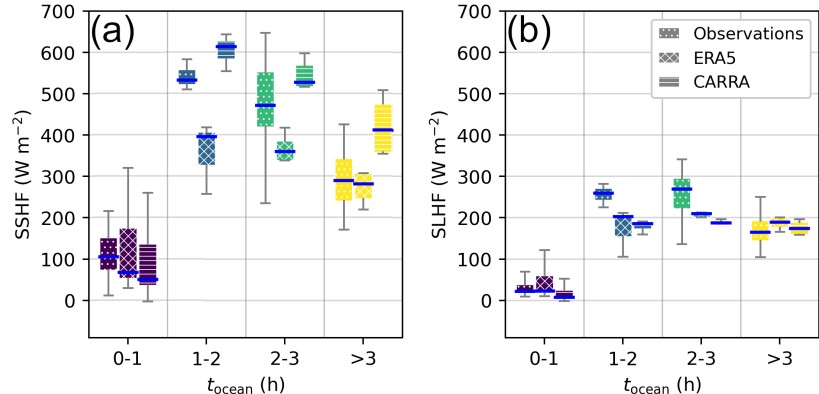

**Figure 7.** a) Surface sensible heat fluxes (SSHFs) and b) surface latent heat fluxes (SLHFs) as derived from observations, ERA5 and CARRA. Box plots show the median as thick lines, the 25-75 quantiles as boxes, as well as the 5-95 quantiles indicated as whiskers. Data is grouped by time over ice-free ocean.

Net diabatic temperature changes in the atmospheric column can result from radiative and cloud (evaporation/condensation) processes, and others. In the case of MCAOs, the primary source for sensible heat is the warm ocean surface (Pithan et al.,

2018). To study the evolution of the heating profiles along the MCAO, it is thus essential to investigate the surface sensible heat

fluxes (SSHFs). A comparison between the observation-derived and reanalysis-based SSHFs is given in Figure 7a. All data are

grouped by time spent over ice-free ocean. Over the Arctic sea ice, SSHFs are already relatively high, with values peaking

above 200 W m$^{-2}$. Both reanalyses show a slightly too low median. This might be caused by the missing sea-ice leads, which

can locally lead to substantial SSHFs (Li et al., 2020). Still, observations and reanalyses are quite close to each other, so the

necessary parameterizations seem to work satisfactorily to allow sufficient heat to escape the ocean upwards.

After the air masses cross the MIZ, large values of SSHF of around 550 W m$^{-2}$ are observed. ERA5 significantly underestimates this value by at least 150 W m$^{-2}$, while CARRA slightly exaggerates it. At 2-3 h into the MCAO, the observed SSHFs

drop slightly, yet are again best captured by CARRA. Only at the later stage are the CARRA SSHFs too high, exceeding

observations by about 100 W m$^{-2}$. In general, the reduction of SSHFs over time is expected, as the temperature difference be-



tween the sea surface and the overlying air is reducing. This can potentially be counterbalanced e.g. by increasing underlying sea-surface temperatures, increased winds, or decreased surface roughness (Papritz and Spengler, 2017).

The different parameters that are required for the calculations of SSHF as shown in Equation 1 are investigated in the supplemental Figure A6. Notably, over ocean both reanalyses significantly under-estimate $U_{10m}$, with CARRA being always closer to the observations. The horizontal thermal gradient between sea ice and the ice-free water surface cause a marked off-

ice breeze, an analogue to sea-land breezes. Similar as reported by Brümmer (1996), in our case, $U_{10m}$ reached its maximum near the ice edge, and the off-ice acceleration due to thermal contrasts is estimated to be around $2.6 \, \mathrm{m \, s^{-1} \, h^{-1}}$ (calculations can be found in Appendix B). Therefore, the $U_{10m}$ in CARRA might be closer to observations than ERA5 as (i) the MIZ is thinner in CARRA, and (ii) the discussed near-surface warm bias over sea ice is weaker in CARRA. Previous studies have also found ERA5 underestimating highest near-surface winds over the ocean next to the MIZ, as well as SSHFs and SLHFs over the MIZ

(Renfrew et al., 2021). Feeding coarse-resolution sea-ice data (with a MIZ of around 80 km, such as in ERA5) into higher resolution models was also found to smear out the simulated fluxes, as well as the rapid increases in air temperature, wind speed, and surface fluxes (Renfrew et al., 2021).

In order to check whether the differences between CARRA and ERA5 discovered for 01 April 2022 are of systematic nature, a climatological comparison of SSHFs from both reanalyses for 1991-2022 can be found in Figure A8. It shows that during

MCAO conditions, CARRA SSHFs are systematically larger than ERA5 SSHFs, and this is consistent over several decades. It is especially pronounced over ocean and corroborates our results for the case study 01 April 2022. Similar systematic differences in the output surface turbulent heat fluxes have been reported also for comparisons of other reanalyses (Zhang et al., 2016; Taylor et al., 2018). Underestimated fluxes result in too low uptake rates for heat and moisture, particularly close to the ice edge (Tomassini et al., 2017; Spensberger and Spengler, 2021). However, similar studies like Slättberg et al. (2023)

are required for a deeper systematic evaluation of ERA5 versus CARRA, for example to disentangle the combined effects on $U_{10m}$ as caused by MIZ width, parameterized surface roughness, or synoptic patterns.





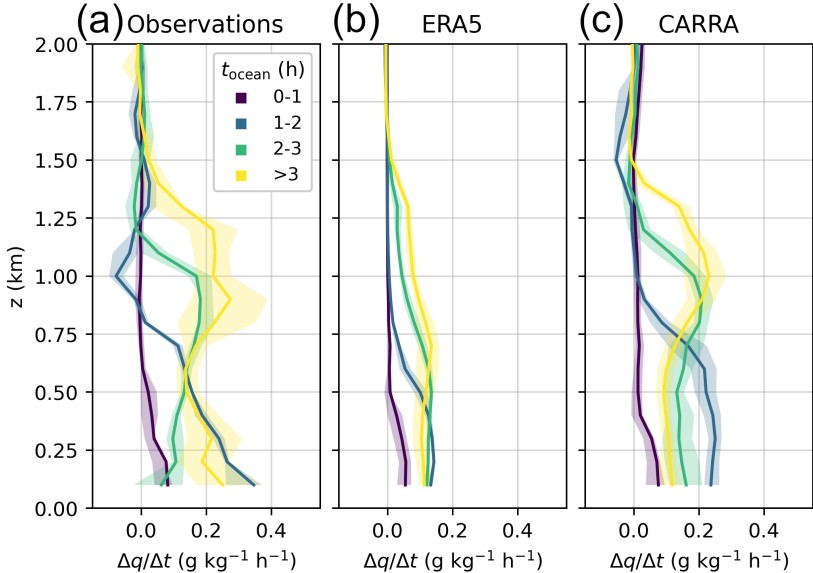

**Figure 8.** Moistening rates expressed as change of specific humidity $q$ per hour as a function of time air masses spent over the ice-free ocean. Lines depict the mean values, shading the 25-75 quantiles. a) Moistening rates based on the quasi-Lagrangian dropsonde observations. b) Corresponding moistening rates extracted from ERA5. c) Corresponding moistening rates extracted from CARRA.

Figure 8a shows the vertically resolved moistening rates based on the quasi-Lagrangian dropsonde observations. For the air masses mostly sampled over sea ice, only minimal moisture uptake is found. This might indicate that some of leads are already frozen over, allowing sensible but not sufficient latent heat to propagate into the atmosphere. The highest uptake at $t_{\mathrm{ocean}} = 1\text{-}2\,\mathrm{h}$ reaches around $0.4\,\mathrm{g\,kg^{-1}\,h^{-1}}$at the surface. For longer times over the ice-free ocean, this moisture is then quickly mixed upwards. The magnitude of upward mixing partly exceeds the moisture uptake near surface at later stages.

Figure 8b shows the corresponding rates as extracted from ERA5. ERA5 underestimates the near-surface moistening rates significantly; also layers further up show rates which are 2-3 times too low. CARRA performs better than ERA5 (Figure 8c). Not only are the near-surface moistening rates closer to observations, but also the upward mixing is more realistic. An insufficient moistening rate within the lower troposphere during a MCAO can be caused by (i) an insufficient supply of moisture from the surface, i.e., too-low SLHF, and/or (ii) an exaggerated removal of water vapor from the atmospheric column. Here, we check both factors separately.

Figure 7b compares the SLHFs between observations and the two reanalyses. Over sea ice, very low SLHFs with a median below $25\,\mathrm{W\,m^{-2}}$ are found. Over the ocean, both ERA5 and CARRA underestimate the SLHF, and they are close to each other. Surprisingly, ERA5 always predicts slightly higher SLHFs than CARRA, which at first does not fit with the much lower moistening rates. The climatological comparison under MCAO conditions shows that ERA5 SLHFs exhibit a constant bias towards larger values than in CARRA (Figure A8). Overall, these findings hint towards mechanisms in ERA5 leading to an exaggerated removal of water vapor, namely cloud processes and precipitation. This is investigated in the next section.




### 3.3.3 Cloud properties

To help understand possible errors in the reanalyses connected to cloud physics, in situ Nevzorov measurements of cloud liquid water and ice contents by the Polar 6 aircraft are utilized (Lucke et al., 2022). A deeper investigation of cloud microphysical processes, such as riming and precipitation formation, is outside the scope of this article; however, details on these processes specifically including the MCAO on 01 April 2022 are reported by Schirmacher et al. (2023) and Maherndl et al. (2023).

As Polar 6 has a range much lower than HALO, it was able to only sample the first 3 h of the MCAO. Figure 9 shows the
height-resolved measured specific cloud liquid water contents $q_{liq}$. Over the closed sea ice (Figure 9a), in the sampled lowest 0.6 km above ground, no cloud liquid water was found, yet with low amounts at the top of the ABLs. After the drift across the MIZ, noticeable amounts of cloud water of up to 0.07 g kg$^{-1}$ are seen up to around 1 km altitude, which corresponds to the altitude of moisture uptake. Surprisingly, the liquid water content decreases in the next time step. This might be correlated with an increase of the frozen hydrometeors (i.e., cloud ice+snow) depicted in Figure A7a. Several in situ probes confirm the
occurrence of riming during the flight of Polar 6 (Maherndl et al. (2023)). With the air temperatures always in the range of -25 °C to 0 °C (see Figure 3), mixed-phase clouds are possible, and also the typical pattern of a supercooled layer above the ice layer is reproduced.

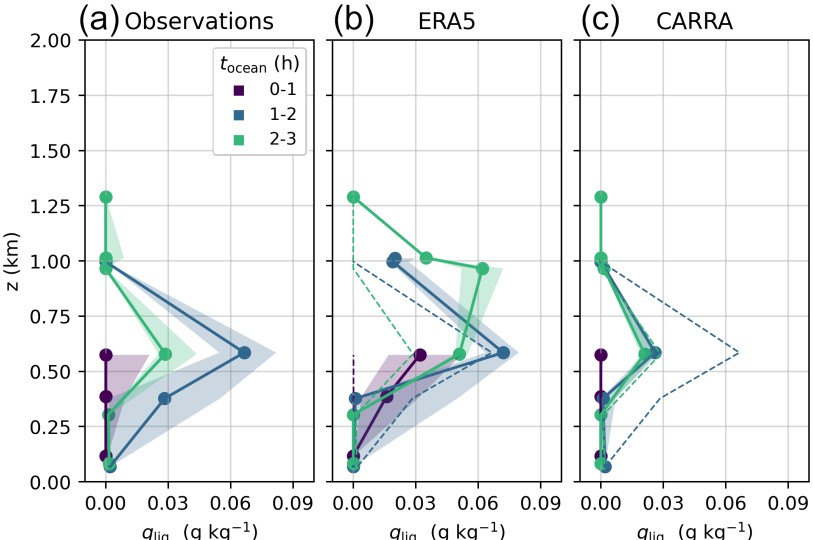

**Figure 9.** Vertical profiles of cloud liquid water content, grouped by time air masses spent over ice-free ocean. a) Observations from a Nevzorov sonde aboard Polar 6, b) taken from ERA5, with the observed values as dashed lines, and c) taken from CARRA, with the observed values as dashed lines.

Figure 9b shows the cloud structures as reproduced by ERA5. In all MCAO stages, ERA5 significantly over-estimates the amount of liquid water present in the clouds. A similar enhanced abundance of liquid-bearing clouds especially over sea ice
has been reported for the IFS, the model behind ERA5 (Tjernström et al., 2021; McCusker et al., 2023). In the MCAO case





here, this is in contrast to CARRA (Figure 9c). With the exception of missing the strong increase in liquid clouds at $t_{\mathrm{ocean}} = 0$-1 h, CARRA matches the observations very well. As a result, total precipitation at the surface is much higher in ERA5 than in CARRA, which creates an additional sink for atmospheric moisture already over sea ice (Figure 10). The statistical comparison between ERA5 and CARRA presented in Figure A8 substantiates the finding of this case study. Figure A8c shows the strong

tendency of ERA5 to form liquid-bearing clouds already over sea ice and the MIZ. Over the ocean, there is a strong bias towards higher cloud liquid hydrometeor contents. The ERA5 clouds systematically precipitate stronger over the MIZ and ocean than clouds in CARRA (Figure A8e). The significance of our findings is reinforced by McCusker et al. (2023), who showed that issues such as an over-abundance of low, liquid-bearing clouds can propagate into higher-resolution models through large-scale forcings.

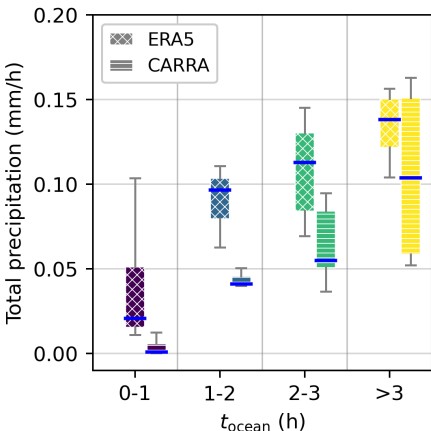

**Figure 10.** Total precipitation reaching surface level for ERA5 and CARRA. Box plots show the median as thick lines, the 25-75 quantiles as boxes, as well as the 5-95 quantiles as whiskers. Data is grouped by time over ice-free ocean.

## 4   Summary and conclusions

We have presented a combined Eulerian and quasi-Lagrangian analysis of an Arctic marine cold air outbreak (MCAO). The MCAO was closely sampled as part of the HALO-(AC)[3] airborne campaign on 01 April 2022 in Fram Strait, west of Svalbard. It was a representative MCAO with common intensity, which can be considered typical for this location and time of year. The performance of two state-of-the-art atmospheric reanalyses, ERA5 and CARRA, was evaluated against the measurements,

with a focus on thermodynamic (temperature, humidity) and cloud properties (cloud liquid water content). We furthermore apply the quasi-Lagrangian approach to convert observations from both HALO and Polar 6 into a common coordinate system ("time over ocean"). The spatio-temporally highly resolved airborne measurements allow for a thorough characterization of the state of the lower troposphere over Arctic sea ice, the marginal sea-ice zone (MIZ), and the ice-free ocean. To the best of our knowledge, we are the first to report on height-resolved diabatic heating and moistening rates in a developing MCAO directly




derived from quasi-Lagrangian observations. Going back to the research questions posed at the beginning of this article, we can answer them as follows.

(Q1) How do air temperature, specific humidity, and clouds evolve in the first four hours of the developing MCAO? Still over sea ice, some leads cause a weak heating and moisture uptake into the shallow atmospheric boundary layer of around 0.2 km height. Within the first hour of departing the closed sea ice, the strong contrast between the cold and dry Arctic air masses and the much warmer ocean cause a diabatic heating larger than 6 K h$^{-1}$ at the surface, a moisture uptake of more than 0.3 g kg$^{-1}$ h$^{-1}$, and the formation of mixed-phase clouds. As time progresses and clouds start forming, heat and moisture mix upwards vertically in the developing marine boundary layer. After four hours, the atmospheric boundary-layer height exceeds 1.5 km. At around the boundary-layer heights, a slight net diabatic cooling and moisture loss are registered, which can be attributed to the upward mixing of air masses into the original, overlying warmer inversion.

(Q2) How do the ERA5 and CARRA reanalyses perform with respect to observations, and compared to each other? In the Eulerian (i.e., fixed in space) framework, the coarse-resolution ERA5 reproduces some well-known issues. The skin and near-surface air temperatures are exaggerated, atmospheric boundary-layer heights are too large, and too many clouds are present. CARRA significantly improves all of these issues: For air temperature over sea ice, ERA5 features a mean absolute error (MAE) 60 % higher than CARRA (1.4 K versus 0.9 K), while for specific humidity over ice-free ocean the MAE is found to be 70 % higher in ERA5 compared to CARRA (0.12 g kg$^{-1}$ versus 0.07 g kg$^{-1}$). Taking the quasi-Lagrangian perspective, the heating rates are reasonably reproduced both in ERA5 and CARRA. However, the strong initial surface-based heating is not captured by ERA5. Even more pronounced are the differences in the moistening rates, where ERA5 estimates are up to 30 % too low, and much better captured by CARRA. Overall, our height-resolved diabatic heating and moistening rates extend the quasi-Lagrangian, ERA5-based climatological MCAO investigation of Papritz and Spengler (2017) to the third dimension. However, as the intense fluxes and transformations in the MIZ are not represented well by ERA5, the heating and especially moistening rates reported by them are likely biased towards lower values.

(Q3) What are some possible sources of errors, which could explain deviations between reanalysis output and observations? Generally, uncertainties in reanalyses can stem from insufficient spatiotemporal resolutions, different measurement sets being assimilated, and also the underlying model physics. The observed discrepancies between the two reanalyses and the obser-vations result from the complex interplay of several processes. Over sea ice, the missing snow on ice layer leads to skin and near-surface air temperatures being too high in ERA5, which might explain the exaggerated boundary-layer heights. Moreover, it is well established that the MIZ is too wide in ERA5. Thus, turbulent fluxes are underestimated significantly in the the first two to three hours of the MCAO. The reduced 10 m wind speeds might be related to the too wide MIZ as well. Especially for the surface sensible heat flux, CARRA almost completely fixes this. ERA5 forms liquid-bearing clouds too early and too thick. This can either be as initial temperatures are slightly too warm, due to issues with parameterization of the mixed-phase clouds, or a combination of both. In all stages investigated, ERA5 clouds thus precipitate considerably more than in CARRA, and too much water vapor is lost to this sink. A similar propagation of errors in initial conditions has been previously reported to affect the atmospheric state hundreds of kilometers downstream.

Overall, we find CARRA fulfilling its intended goal of improving on the global ERA5 reanalysis with regard to the ther-
modynamic and cloud evolution, based on the parameters investigated in the critical first four hours of the MCAO. CARRA
might thus be better suited for driving higher-resolution models, such as large eddy simulations. While some climatological
comparisons of differences between ERA5 and CARRA were supplied, deeper investigations are required to further support
the statistical significance of our findings, and to determine which components of CARRA are primarily responsible for the
improvements. Ideally, these analyses should include extended data rows of observations, such as from regular radiosonde
launches. Finally, the unprecedented quasi-Lagrangian observations collected during HALO-(AC)[3] pose a rich database for fu-
ture studies. For example, sensitivity studies could reveal the influence that initial aerosol concentrations (cloud-condensation
nuclei, ice-nucleating particles) and different cloud schemes (one-moment or two-moment) have on the vertical mixing of heat
and moisture, especially considering the intense surface forcings.

*Data availability.* Most airborne observational data used in this study was accessed through the ac3airborne module (Mech et al., 2022b).
The Nevzorov liquid and total water contents are available from https://doi.pangaea.de/10.1594/PANGAEA.963628. VELOX-derived skin
temperature measurements can be obtained from https://doi.pangaea.de/10.1594/PANGAEA.963401. The CARRA and ERA5 data sets can
be freely retrieved from the Copernicus Climate Change Service (C3S) Climate Data Store (CDS) at https://cds.climate.copernicus.eu/, last
accessed on 5 November 2023. For CARRA, data is available on model levels (Schyberg et al., 2020a), pressure levels (Schyberg et al.,
2020b) and single levels (Schyberg et al., 2020c). Further information can be found in CARRA's documentation (Yang et al., 2023) and user
guide (Nielsen et al., 2023). ERA5 is also available on model levels (Hersbach et al., 2023a), pressure levels (Hersbach et al., 2023b) and
single levels (Hersbach et al., 2023c), see Hersbach et al. (2020). The results presented here contain modified C3S information. Neither the
European Commission nor ECMWF is responsible for any use that may be made of the Copernicus information or data it contains.

*Author contributions.* BK, IS, MK, and MW contributed to conception and design of the study. BK elaborated the methods, performed
the analyses, created the figures, and prepared the original draft. IS and MK supported the development of analysis methods. MS provided
the VELOX-derived skin temperature measurements. NS supported the processing and analysis of CARRA data. MM and JL provided
and discussed the Nevzorov data. HM supported the processing of ERA5 data. All authors discussed the results, contributed to manuscript
revision and approved the final submitted version.

*Competing interests.* The authors declare that no competing interests exist.

*Acknowledgements.* We gratefully acknowledge the funding by the Deutsche Forschungsgemeinschaft (DFG, German Research Founda-
tion) - Projektnummer 268020496 - TRR 172, within the Transregional Collaborative Research Center "ArctiC Amplification: Climate Rel-
evant Atmospheric and SurfaCe Processes, and Feedback Mechanisms (AC)[3]". We are furthermore grateful for funding of project grant no.



316646266 by DFG within the framework of the Priority Programme SPP 1294 to promote research with HALO. The publication of this article was funded by the Open Access Publishing Fund of Leipzig University supported by the German Research Foundation within the program Open Access Publication Funding. We cordially thank the Alfred-Wegener-Institute, German Aerospace Center (DLR), as well as all aircraft crews and participants which made the HALO-(AC)[3] campaign possible. We also thank the Institute of Environmental Physics at the University of Bremen for providing the merged MODIS-AMSR2 sea-ice concentration data (https://data.seaice.uni-bremen.de/modis_amsr2, last access 20-Oct-2023). We acknowledge the use of imagery from the NASA Worldview application (https://worldview.earthdata.nasa.gov/), part of the NASA Earth Observing System Data and Information System (EOSDIS).



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





## Appendix A: Supplemental material

### Calculation of time over ice-free oceans for HALO and Polar 6

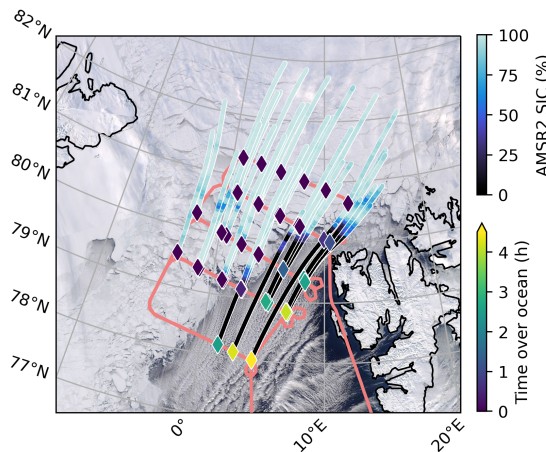

**Figure A1.** Procedure of calculating the time over ice-free ocean for HALO. Air masses are started at 10 hPa above ground for all released dropsonde locations. Trajectories are then calculated 12 h backwards and the AMSR2 sea-ice concentration (SIC; Ludwig et al., 2020) traced. Time is integrated backwards as long as the SIC is below 20 %. The background Terra/MODIS satellite image is taken from NASA Worldview (2023).

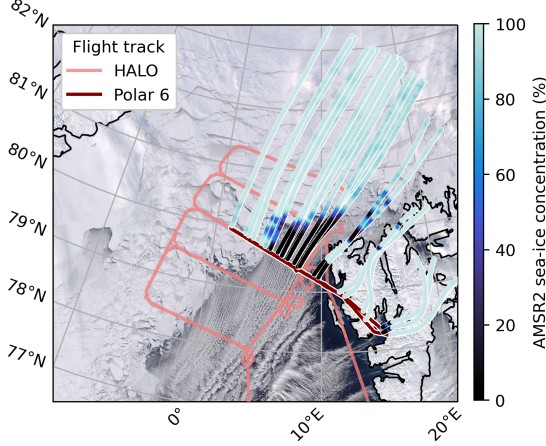

**Figure A2.** Procedure of calculating the time over ice-free ocean for Polar 6. Air masses are every 1 min at the flight level. Trajectories are then calculated 12 h backwards and the AMSR2 sea-ice concentration (SIC; Ludwig et al., 2020) traced. Time is integrated backwards as long as the SIC is below 20 %. The background Terra/MODIS satellite image is taken from NASA Worldview (2023).





**Comparison of wind fields from observations, ERA5, and CARRA**


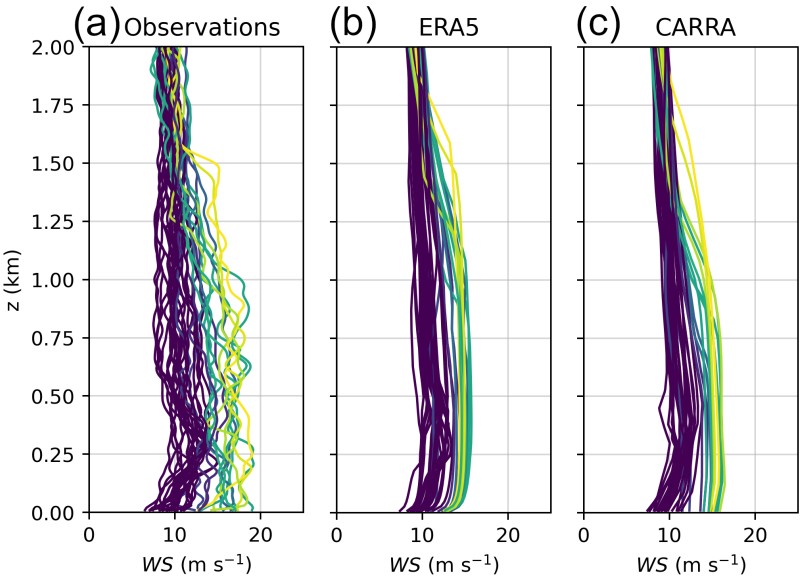

**Figure A3.** Vertical profiles of wind speed (*WS*) taken from a) Observations, b) ERA5, and c) CARRA. All profiles are colored by time air masses spent over ice-free ocean before reaching the dropsonde locations.

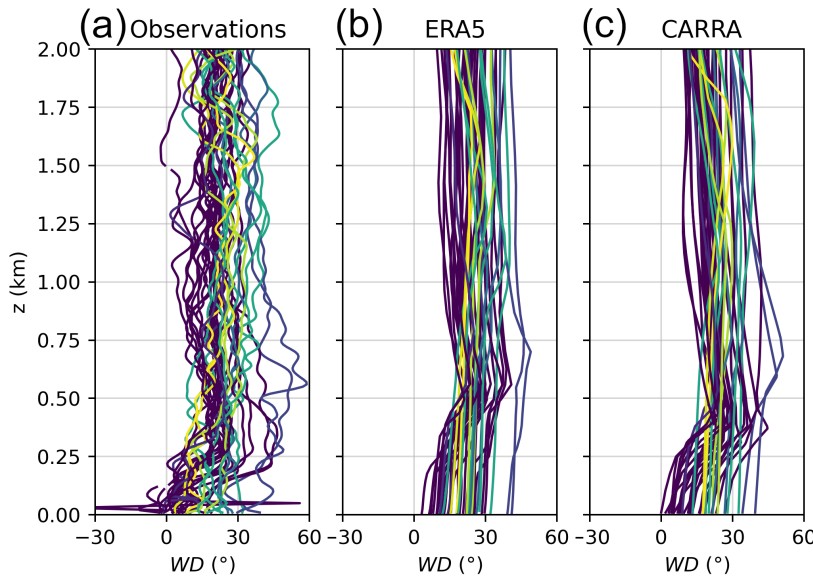

**Figure A4.** Vertical profiles of wind direction (*WD*) taken from a) Observations, b) ERA5, and c) CARRA. All profiles are colored by time air masses spent over ice-free ocean before reaching the dropsonde locations.



**Investigation of possible cloud-top cooling**

The diabatic heating rates not only from observations, but also both ERA5 and CARRA showed altitude ranges where negative values were found, i.e., a net cooling (Figure 6). As these altitudes coincided with the deepening atmospheric boundary layer heights as well as the correlated increasing cloud top heights, it seemed possible that this net diabatic cooling could be a

sign of cloud-top radiative cooling. To test this hypothesis, ERA5 temperature tendencies were analyzed, similar as done by others (You et al., 2021a, b; Kirbus et al., 2023a, b). For this purpose, the temperature tendency due to all processes as well as only by terrestrial radiation were evaluated as function of time air masses spent over open ocean. The result is shown in Figure A5. With increasing time over ice-free ocean, stronger negative diabatic heating rates are found at altitudes close to the BLH (Figure A5a). Yet the air masses don't reside long enough at cloud top to actually experience a significant, net cloud

top cooling due to radiative processes (Figure A5b). Instead, an upward mixing of lower, colder air into the upper, warmer inversion due to turbulent processes seems more likely.

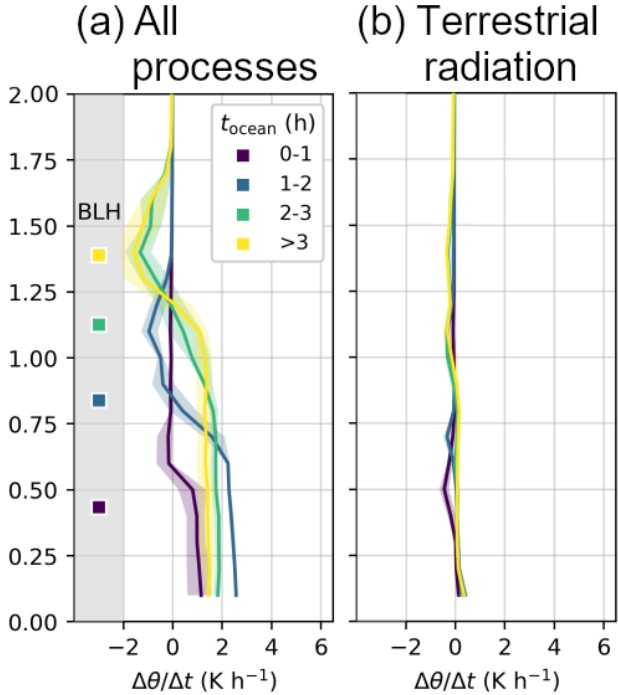

**Figure A5.** Investigation of suspected cloud top cooling. a) Vertical profile of the ERA5 temperature tendency due to all processes, grouped by time air mass spent over open ocean. b) ERA5 temperature tendency due to terrestrial radiation.




**Individual parameters shaping SSHF and SLHF**

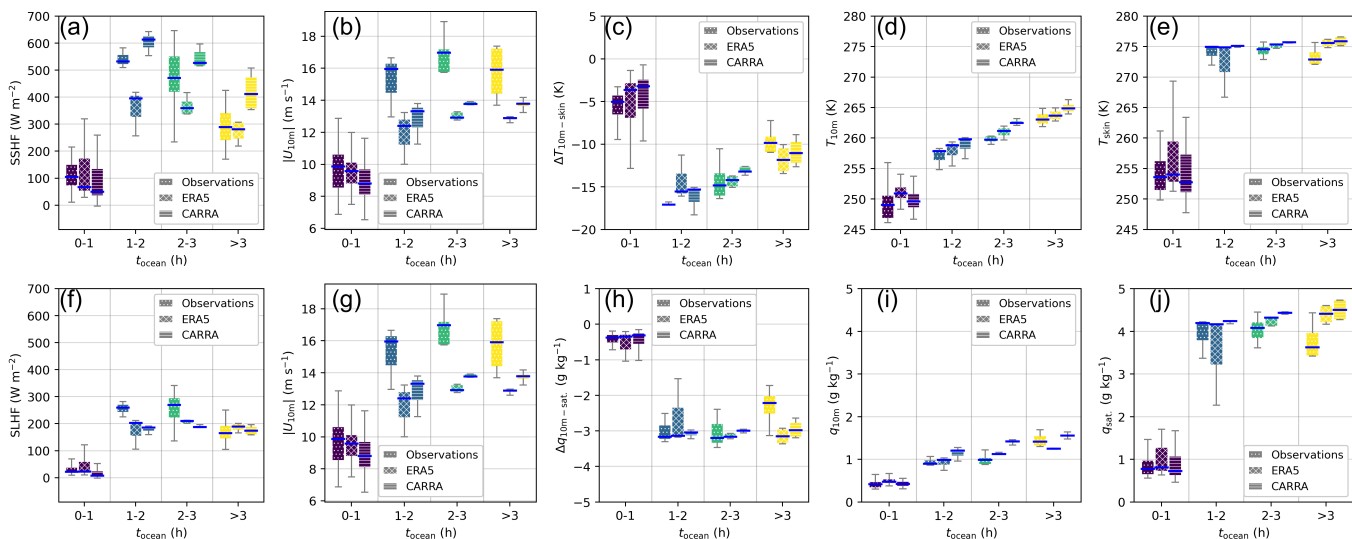

**Figure A6.** Investigation of the different parameters driving surface sensible heat flux (SSHF, panels a-e) and surface latent heaf flux (SLHF, panels f-j). In all panels, data is grouped by the time air masses spent over ice-free ocean, and for observations, ERA5, and CARRA. a) SSHF, b) 10 m wind speed $U_{10m}$, c) temperature difference between 10-m air temperature $T_{10m}$ and skin temperature $T_{skin}$, d) $T_{10m}$, e) $T_{skin}$. f) SLHF, g) $U_{10m}$, h) specific humidity difference between 10-m specific humidity $q_{10m}$ and saturation specific humidity taken at sea-surface temperature $q_{sat}$, i) $q_{10m}$, j) $q_{sat}$.





**Cloud ice+snow water contents**


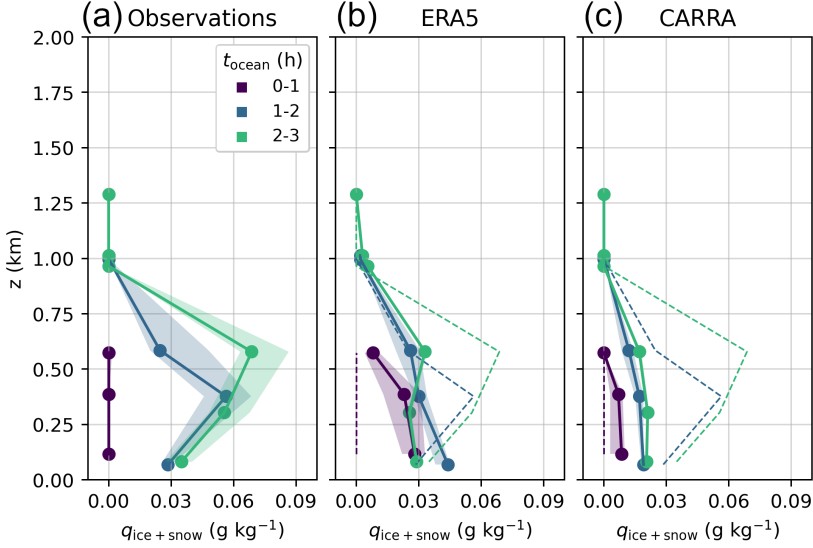

**Figure A7.** Vertical profiles of cloud frozen (ice+snow) water content, grouped by time air masses spent over ice-free ocean. a) Observations from a Nevzorov sonde aboard Polar 6, b) taken from ERA5, with the observed values as dashed lines, and c) taken from CARRA, with the observed values as dashed lines



**Climatological comparison of ERA5 and CARRA**

To check whether the findings on the performance of CARRA and ERA5 on the 01 April 2022 in representing MCAOs are
systematic, the difference between the two reanalyses is investigated on a climatological basis. For all days 1991-2022, the
MCAO index $M_{850hPa}$ is calculated based on ERA5. It is averaged over the Fram Strait box (75-80 °N and 10 °W-10 °E). Only
medium to strong MCAOs (MCAO index larger than 4 K) are retained. For this sub-selection, the surface sensible and latent
heat fluxes as well as diverse cloud-related parameters are investigated (Figure A8). Under MCAO conditions, CARRA shows
systematically higher SSHFs over the MIZ and ice-free ocean, lower SLHFs over all surface types, generally lower cloud liquid
water contents and especially over the ice-free ocean, as well as lower surface precipitation over the MIZ and open ocean.

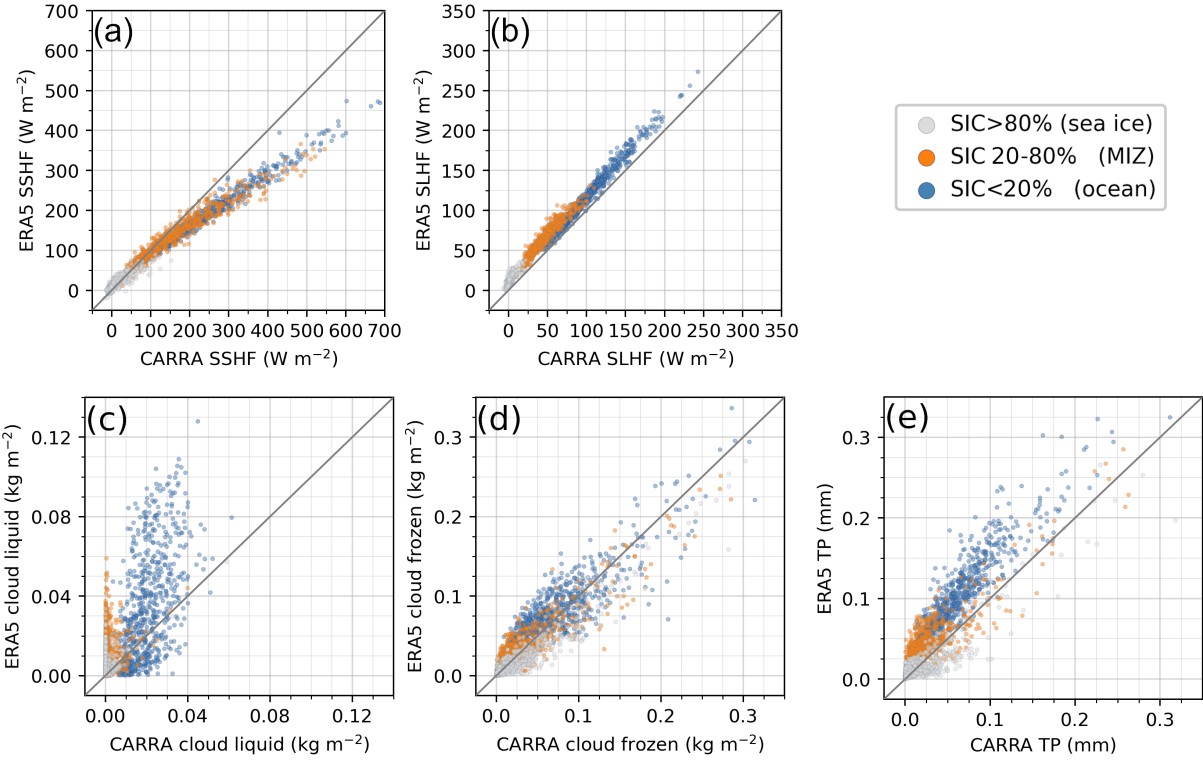

**Figure A8.** Climatological investigation of differences between CARRA and ERA5 MCAOs. All plots depict the daily mean values over the
Fram Strait box, separately for sea ice, the MIZ, and ice-free ocean. Panels show the comparisons of a) SSHFs, b) SLHFs, c) total column
cloud liquid water content, including rain, d) total column cloud frozen hydrometeors, meaning ice+snow, and e) total precipitation (TP).



**Appendix B**

**Estimating the off-ice acceleration**

One key parameter determining SSHF and SLHF is the 10 m wind speed $U_{10\mathrm{m}}$. As was already noted by Brümmer (1996), the horizontal thermal gradient between sea ice and the ice-free water surface cause a marked off-ice breeze, an analogue to sea-land breezes. In our case, $U_{10\mathrm{m}}$ reached its maximum near the ice edge, which has also been observed before by Brümmer (1996). The off-ice breeze can be estimated by starting with the static pressure equation:

$$\Delta p = -\frac{\mathrm{g}\,p}{\mathrm{R}\,T}\Delta z, \tag{B1}$$

where $\Delta p$ denotes the difference in pressure over a $\Delta z$ deep ABL, g the gravitational acceleration, $p$ is the mean pressure, R the gas constant for dry air, and $T$ the air temperature. By differentiating this equation with regard to $T$, we get:

$$\mathrm{d}(\Delta p) = \frac{\mathrm{g}\,p}{\mathrm{R}\,T^2}\Delta z\,\mathrm{d}T \tag{B2}$$

In our case, a BLH of about 200 m thickness is found over sea ice. $T$ shows an increase of 9 K over 100 km. A mean pressure
1010 hPa and mean temperature of 255 K in the ABL are found. From this, a horizontal pressure gradient of around 1 hPa over 100 km results near the surface. While this appears small, the resulting pressure gradient force accelerates air masses in off-ice direction:

$$\alpha = -\frac{1}{\rho}\frac{\mathrm{d}p}{\mathrm{d}x} \tag{B3}$$

On 01 April 2022, this corresponds to an acceleration $\alpha$ of about 2.6 m s$^{-1}$ h$^{-1}$.


**Quasi-Lagrangian approach**

The quasi-Lagrangian analysis conducted for the research flight on 01 April 2022 is performed using *Lagranto* in combination with ERA5 three-dimensional wind fields. Air masses are initialized every 1 min along HALO's flight track, vertically every 5 hPa between 250 hPa and surface, and horizontally evenly spaced every 7 km in a 20 km radius. In total, 2.1 million trajectories
are calculated 6 hours forward in time. Caused by the vertical wind shear of wind directions and wind speed, the sampled air masses start moving in different directions. Only for a certain fraction, due to successful flight planning and/or some luck, some of the same air masses are sampled again in a different location and to a second time. A match is registered if the same air mass is seen again in the column below HALO within the same 20 km radius. In the final step, observations from dropsondes are included. Only those matches in the lowest 2 km are kept where the time difference between the matching air mass below
the aircraft and the dropsonde in its time during descent is below 90 seconds. At a flight speed of around 800 km h$^{-1}$, this again corresponds to a maximum distance of 20 km. A final number of approximately 24,200 matches is thus identified.

