# Peer review of "Thermodynamic and cloud evolution in a cold air outbreak during HALO-(AC)3: Quasi-Lagrangian observations compared to the ERA5 and CARRA reanalyses"

_EGUsphere, 2023_

## Referee Comment (RC2)

Review of manuscript egusphere-2023-2989 submitted to Atmospheric Physics and Chemistry

**Thermodynamic and cloud evolution in a cold air outbreak during HALO-(AC)[3]: Quasi-Lagrangian observations compared to the ERA5 and CARRA reanalyses**

by B. Kirbus et al.

The study evaluates the representation of a marine cold air outbreak (MCAO), i.e., off-ice flow of very cold Arctic air over a much warmer open ocean, in ECMWF's ERA5 and the high resolution CARRA reanalyses based on aircraft and dropsonde observations that were obtained during the HALO-(AC)[3] campaign. The MCAO event under consideration occurred on 1. April 2022 in Fram Strait, a region which is well-known for its frequent and intense MCAOs, where the originally Arctic air is rapidly transformed by heat and moisture uptakes from the ocean. The event was sampled by the HALO and the Polar 6 aircrafts in a quasi-Lagrangian manner providing unique observations of the MCAOs evolution with fetch from sea ice. These observations are used for two purposes: (i) The estimation of diabatic heating and moistening rates as the air masses are transformed over the ocean, and (ii) to evaluate how the two reanalyses represent the MCAO. Kinematic trajectories computed with winds from reanalysis are employed to confirm the Lagrangian matching and to study the air mass transformations in the reanalyses. Profiles of temperature and specific humidity, diabatic heating and moistening rates, surface fluxes, and cloud properties are systematically evaluated first over sea ice and then with increasing fetch from the sea ice edge. Overall, the two reanalyses agree reasonably well with observations, but the authors also find systematic biases, which in part appear to be related to an insufficiently sharp maginal ice zone. In addition, the findings reveal the benefits of higher spatial resolution in CARRA.

MCAOs are the key weather features in which heat and moisture is transferred from the ocean to the atmosphere - they are not restricted to the Arctic but the same physics also occurs at more temperate latitudes in flows across ocean fronts in the storm track entrance regions. Hence, it is of great importance that these processes are well represented in models and reanalyses. This study, thus, fills an important gap.

I truly enjoyed reading the manuscript and find the results significant. I have no doubt that the study is of great interest to the readership of ACP. Overall, the analyses appear sound, the manuscript is well written, and the figures are nicely designed and easy to read. I have a few suggestions for improvement, which are detailed below, which are all rather minor in nature. Once these points have been addressed, I recommend the manuscript be accepted for publication.

**Specific comments:**
1. Detailed observations of MCAOs, in particular also in the Nordic Seas, have been obtained in earlier field campaigns, documenting in particular the boundary layer evolution, heat and moisture budgets, as well as estimates of surface fluxes (e.g., Shapiro et al. 1987, Brümmer 1997, 1999, Vihma and Brümmer 2002...; detailed references are

given below). I think that these studies should be more extensively discussed in the introduction.

2. L30: Please explain what you mean by *decoupling*.

3. L45: Svingen et al. (2023) provide observational evidence for the importance of MCAOs for deep water formation.

4. L70ff: You raise an important point here. Kinematic trajectories only see the grid-scale winds but they are not aware of the sub-gridscale (turbulent) motions. As a result, a kinematic trajectory represents a volume of air floating with the mean winds with turbulent exchanges happening through its boundaries and appearing as sources / sinks. Perhaps this point could be made a bit more transparent? This is in contrast to the observed profiles, which in some sense suffer from the opposite problem: they capture all the small scale fluctuations, which are not necessarily representative of the mean.

5. L125ff: I am wondering how the transfer coefficients are computed. In general they are modeled as a function of static stability and surface properties such as the roughness length. How do you compute the transfer coefficients?

6. Section 2.3: I think it would help if the additional explanations of the quasi-Lagrangian approach given in the appendix B would be included here in the main part.

7. Section 3.2.2: Does the cold bias in the profiles over sea ice go along with a too deep cold BL? And related to that, how does the mean temperature across the entire BL compare to observations? As I understand, the mean biases presented in Table 1 are computed using the observed BLH not the BLH as represented in the reanalyses. Given the dipole structure of the biases in the vertical profiles, the mean biases will critically depend on how you define the BLH.

8. Section 3.3.2: The diabatic heating and moistening profiles are very interesting. How do the differences between observations and reanalyses fit together with the differences in surface fluxes? For example, at the later stages of the MCAO evolution, the diabatic heating profiles in ERA5 show that the warming extends too far up. This suggests, that overall there is too much energy input into the BL, which hin turn seems inconsistent with an underestimation of the surface sensible heat fluxes.

9. L408ff: I can't fully reconcile the statement that ERA5 overestimates cloud liquid water content at all stages. For the time range 1-2h the values shown in Fig. 9b seem to agree quite well with observations, or am I missing something?

10. Generally, I feel that some of the figures presented in the Appendix are interesting but not key to the understanding of the study (specifically figures A3 - A8). Hence, I suggest moving them from the Appendix to the Supplement along with the corresponding text.

**Editorial comments**

l2: *intensive -> intense*

l8: delete *specific* in *investigated specific MCAO*

l11: Suggest to rephrase *As the air mass continued its drift southwards, ...* as *With increasing fetch of the sea ice edge, ...*

l12: *quasi-Lagrange -> quasi-Lagrangian*

l16: *... issues with **the representation** of ...*

l63: ***at** high temporal resolution*

l99: *as **a** dedicated*

l108: *several  such flight legs*

l110: *reported -> performed?*

l120: vertical gradient (?)

l125: Please fix the reference ECMWF, 2016 by including the appropriate authors

l154: *The reanalysis data can be retrieved for...*

l189: Please spell out LAGRANTO (Lagrangian Analysis Tool)

l237: replace *obvious* by *evident*

l241: *liquid+ice ->* liquid and ice

l245: replace *issue* by *matter*

l246: *the sea ice concentrations **in** ERA5 **are***

l274: please rephrase *respective observations derived BLH*

l294: *... vertical velocity is used **for** the three-dimensional...*

l404: here and elsewhere: *cloud ice **and** snow*

l405: fix ref to Maherndl et al (2023)

l423: *common intensity* – Do you mean *typical intensity*?

L453: *resolution* (singular)

Caption Fig. 2: *... diamond shapes show the location**s** of released ...*

Caption Fig. 3: *... indicated **on** the left hand side ...* (please also fix in other captions)

**Literature:**

Brümmer, B., 1997: Boundary layer mass, water, and heat budgets in wintertime cold-air outbreaks from the Arctic sea ice. Mon. Wea. Rev., 125, 1824–1837

Brümmer, B., 1999: Roll and cell convection in wintertime Arctic cold-air outbreaks. J. Atmos. Sci., 56, 2613–2636

Shapiro, M. A., L. S. Fedor, and T. Hampel, 1987: Research aircraft measurements of a polar low over the Norwegian Sea. Tellus, 39A, 272–306

Svingen, K., A. Brakstad, K. Våge, W. von Appen, and L. Papritz, 2023: The Impact of Cold-Air Outbreaks and Oceanic Lateral Fluxes on Dense-Water Formation in the Greenland Sea from a 10-Year Moored Record (1999–2009). *J. Phys. Oceanogr.*, **53**, 1499–1517

Vihma, T., and B. Brümmer, 2002: Observations and modelling of the on-ice and off-ice air flow over the Northern Baltic Sea. Bound.-Layer Meteor., 103, 1–27

---

## Author Comment (AC1)

**Thermodynamic and cloud evolution in a cold air outbreak during HALO-(AC)³: Quasi-Lagrangian observations compared to the ERA5 and CARRA reanalyses**

Ref.: EGUSPHERE-2023-2989
Atmospheric Chemistry and Physics (ACP)

Dear editor, dear anonymous reviewers,

We would like to express our gratitude for the thoughtful comments towards our submission. We have carefully considered all suggestions and comments raised. In the following paragraphs, you will find a detailed reply to each point. We are confident that by addressing the comments and suggestions, we have been able to further improve and refine our manuscript.

Best regards, on behalf of all co-authors

Benjamin Kirbus

**Referee #1**

One criticism is that I found the cloud discussion within the paper lacking and gave me a feeling that it was added at the end. The cloud results warrant further consideration.

- o Thank you for this comment. To give more details on the clouds, we added the additional panel Figure 9a, which adds the radar perspective from aboard HALO.
- o The shown radar reflectivities are also used in the discussion of precipitation and reinforce the notion that ERA5 clouds precipitate too early.
- o However, we prefer to keep the discussion of cloud vertical profiles at the end of the manuscript. The thermodynamic evolution is the key point of the article (and thus also placed first in the title), and the cloud evolution is mostly used to explain the moisture uptake rates.

The only major comment is the need for a more thorough presentation and discussion of the errors in the measurements and analysis throughout the manuscript. The authors do a nice job performing analysis that tests the impact of assumptions and uncertainties (e.g., performing trajectory analysis with both ERA5 and CARRA). However, the uncertainties are not given in a quantified manner. The uncertainty in the surface turbulent fluxes determined using the bulk formula and the differences using ERA5 and CARRA for trajectory analysis in terms of spatial differences (parcel trajectories where 3 km different at 4 hours) would help the readers assess the confidence in the results.

- o We agree that readers should have a better overview of the uncertainties connected to the trajectory calculations and to the estimated turbulent fluxes.
- o In the figures of diabatic heating rate and moistening rate as deduced from observations (Figures 6 and 8), we added dashed lines. These dashed lines represent the observation-derived rates when using CARRA wind fields instead of ERA5 wind fields (solid lines). We also calculate and discuss the mean absolute errors on $\Delta\theta/\Delta t$ and $\Delta q/\Delta t$ in the lowest 1.5 km when switching from ERA5 to CARRA trajectories. Finally, we adjusted the Methods and figure captions to reflect the new approach.
- o Concerning the observation-derived surface turbulent fluxes, we decided to adjust our calculation method. So far, we had used the same parametrizations as in ERA5, and used the same transfer coefficients as in ERA5; however, this approach was not sound, as the underlying parameters such as $T_{10m}$ etc. differ between measurements and ERA5. To calculate

the surface turbulent fluxes, we now employ the COARE 3.5 bulk air-sea flux parametrization (Fairall et al., 2003). This is also used for the calculation of transfer coefficients. COARE 3.5 is a widely used algorithm, and it is independent from ERA5. In the wind speed range up to 20 m s$^{-1}$, it has been reported to exhibit uncertainties of approximately 10% of the turbulent fluxes. Together with the uncertainties stemming from our airborne measurements, we assume a total uncertainty of around 12% of the turbulent fluxes. All of this is now discussed in the updated manuscript. Furthermore, statements are added to highlight the handling and increased uncertainty of turbulent flux calculations in the MIZ.

- o Finally, we want to highlight the already existing shading of the 25-75th quantiles in the plots of diabetic heating and moistening rates. These deliver additional estimates for uncertainty from the sampling error itself (i.e., the different dropsondes contained in each category).

L145: "137 model levels" doesn't provide the vertical resolution? Please state the vertical resolution in the boundary layer for ERA5, as this will influence the model representation. L157: same comment on CARRA vertical model levels.

- o That is correct, and was not very clear in the first draft. In ERA5, model levels start 10m above ground, and are then placed approx. every 20m, with an increasing spacing upwards. In CARRA, model levels start 15m above ground, then approx. every 30m, again increasing upwards. The updated manuscript now details this.

L 178: I suggest changing the word "hints" to "indicates" or "suggests."

- o We followed your suggestion and changed the wording to "indicates".

L245-255: It seems that the statement "The sharper MIZ of CARRA in comparison to ERA5 is not only and issue of spatial resolution…" is not completed. I'm left confused as to what the difference is. The sea ice data set used for ERA5 is discussed but no sea ice data for CARRA is described here. Please clarify what the source of the difference in the MIZ sharpness between CARRA and ERA5.

- o Thank you for pointing this out. Indeed, in the first version of the manuscript, information on CARRA's sea ice input was lacking. CARRA heavily relies on the European Space Agency's Climate Change Initiative product (SICCI), which incorporates AMSR-2 satellite data at a resolution of approx. 15 km. The output fields of SICCI are on a 25 km scale, but CARRA uses complex processing together with higher-resolution sea surface temperature data to achieve its improved sea ice fields. This is now explained in the updated manuscript.

L272-276: Are the MAEs in ABL temperature and specific humidity pressure-weighted?

- o No pressure-weighting was used, as the ERA5/CARRA data on model levels was first interpolated to a regular vertical coordinate of altitude above ground (m). We clarified the processing of MAEs by introducing one sentence (shown in italics):
  "The mean absolute errors (MAEs) of ERA5 and CARRA with regard to measurements are computed. *Output from both reanalyses as well as dropsondes is interpolated to a common vertical coordinate of altitude above ground in 10m steps.* To evaluate especially the crucial ABL representation, …".

L311: What is meant by "weigh matches by their frequency of occurrence"?

- o We rephrased the sentence to "Finally, the approach presented here of initializing and then registering matches for a large number of  trajectories within a radius of 20km around HALO's location is also *essential to better assess the statistical significance of matches*".

L348: A 200 W m$^{-2}$ sensible heat flux strikes me as a bit high and was unexpected. It is certainly substantial. Can you influence the surface-air temperature difference that goes along with this flux and provide an uncertainty estimate for the SSHF and SLHF values? These values are included in the appendix figure (A6) and it would be good to reference specific values within the text to provide the reader additional context.

- As detailed above, we now switched to the COARE 3.5 bulk air-sea flux algorithm, which is independent from ERA5, and incorporates all the observations from the dropsonde. It is reported to lie within approximately 10% of the real turbulent fluxes for open oceanic waters. Considering our measurement uncertainties, this increases to at least 12%. We detail this also in the text.
- The new SSHF in the MIZ (time over ocean 0-1 h) now lies lower, at a median of below 50 W m$^{-2}$, 25-75th quantiles at 0-80 W m$^{-2}$, 5-95th quantiles at 0-150 W m$^{-2}$. However, we added additional text highlighting the much larger uncertainties especially in the MIZ, where assumptions on the lead fraction must be included. Also, we do not model the fluxes over the sea ice itself, assuming them to be much smaller in comparison.

L378: "This might indicate that some leads are already frozen over…" I'm confused by this statement since the bulk formula used in the calculations doesn't include information on the lead fraction. The bulk formula used only has a surface-air specific humidity difference. Thus, the formulation used wouldn't have a direct sensitivity to lead fraction (e.g., different transfer coefficient over ice and ice-free ocean. If this is the case, this statement would be incorrect. Please explain/revise.

- Indeed, this sentence was not formulated well and contained speculation. We removed it, and instead a) described the handling of the COARE 3.5 air-sea flux algorithm over the MIZ, where we use the AMSR2 satellite data to roughly estimate open water fraction, and importantly also b) highlighted the much larger uncertainties of SSHF and SLHF calculations over the MIZ.

**Cited literature**

Fairall, C. W., Bradley, E. F., Hare, J., Grachev, A. A., and Edson, J. B.: Bulk parameterization of air–sea fluxes: Updates and verification for the COARE algorithm, Journal of climate, 16, 571–591, https://doi.org/10.1175/1520-0442(2003)016<0571:BPOASF>2.0.CO;2, 2003.

**Thermodynamic and cloud evolution in a cold air outbreak during HALO-(AC)[3]: Quasi-Lagrangian observations compared to the ERA5 and CARRA reanalyses**

Ref.: EGUSPHERE-2023-2989
Atmospheric Chemistry and Physics (ACP)

**Referee #2**

Detailed observations of MCAOs, in particular also in the Nordic Seas, have been obtained in earlier field campaigns, documenting in particular the boundary layer evolution, heat and moisture budgets, as well as estimates of surface fluxes (e.g., Shapiro et al. 1987, Brümmer 1997, 1999, Vihma and Brümmer 2002...; detailed references are given below). I think that these studies should be more extensively discussed in the introduction.

- o Thank you for hinting towards these publications, which we all incorporated into the manuscript.
- o We cite Vihma and Brümmer (2002) with regards to mesoscale modeling of CAOs.
- o We cite Shapiro et al. (1987) with regards to airborne CAO studies, as well as the magnitude of SSHF+SLHF in CAOs.
- o Most importantly, we included Brümmer et al. (1997, 1999) with regards to airborne CAO studies and the importance of entrainment fluxes. In the early stages of CAOs, the entrainment fluxes (through mixing of colder air below with warmer air aloft from the original Arctic inversion) can grow to reach 30-80% of the surface sensible heat fluxes. We also use the details reported by Brümmer et al. later in the discussion of ERA5 vs. CARRA vertical heating rates, see our comment further below.

L30: Please explain what you mean by decoupling.

- o We clarified the term to "*atmospheric boundary layer decoupling*".

L45: Svingen et al. (2023) provide observational evidence for the importance of MCAOs for deep water formation

- o Thank you very much for pointing out this recent publication, which provides actual observations to back up the claimed correlation. We added Svingen et al. to the literature cited in L45.

L70ff: You raise an important point here. Kinematic trajectories only see the grid-scale winds but they are not aware of the sub-gridscale (turbulent) motions. As a result, a kinematic trajectory represents a volume of air floating with the mean winds with turbulent exchanges happening through its boundaries and appearing as sources / sinks. Perhaps this point could be made a bit more transparent? This is in contrast to the observed profiles, which in some sense suffer from the opposite problem: they capture all the small scale fluctuations, which are not necessarily representative of the mean.

- o We agree that it is best to be transparent in the opportunities, but also challenges connected to kinematic trajectory calculations. We adjusted the text to reflect this. It now reads:
  "Instead, wind fields  as available from reanalyses are used to model the flow of air masses (Sprenger and Wernli, 2015). *Such kinematic trajectories are oblivious to sub-gridscale turbulent motion leading to exchanges across neighboring air masses. Yet they account for the mean drift along prevailing winds, and they allow for* example aircraft to be  employed to trace specific air mass parcels along their trajectory (...)".

L125ff: I am wondering how the transfer coefficients are computed. In general they are modeled as a function of static stability and surface properties such as the roughness length. How do you compute the transfer coefficients?

- o Concerning the observation-derived surface turbulent fluxes, we decided to adjust our calculation method. So far, we had used the same parametrizations as in ERA5, and used the same transfer coefficients as in ERA5;  however, this approach was not sound, as the underlying parameters such as $T_{10m}$ etc. differ between measurements and ERA5. To calculate the surface turbulent fluxes, we now employ the COARE 3.5 bulk air-sea flux parametrization. This is also used for the internal calculation of transfer coefficients. COARE 3.5 is a widely used algorithm,  independent from ERA5. In the wind speed range up to 20 m s$^{-1}$, it has been reported to exhibit uncertainties of up to 10% of the turbulent fluxes. Together with the uncertainties stemming from our airborne measurements, we assume a total uncertainty of around 12% of the turbulent fluxes. All of this is now discussed in the updated manuscript.

Section 2.3: I think it would help if the additional explanations of the quasi-Lagrangian approach given in the appendix B would be included here in the main part.

- o We agree that readers unfamiliar with the approach might require a deeper explanation of the quasi-Lagrangian flight strategy. Accordingly, we merged the description of Appendix B with the explanations in Section 2.3

Section 3.2.2: Does the cold bias in the profiles over sea ice go along with a too deep cold BL? And related to that, how does the mean temperature across the entire BL compare to observations? As I understand, the mean biases presented in Table 1 are computed using the observed BLH not the BLH as represented in the reanalyses. Given the dipole structure of the biases in the vertical profiles, the mean biases will critically depend on how you define the BLH

- o Thank you for this good observation. In the original manuscript, we indeed accidentally "cut off" some of the errors especially of the ABL over sea ice, as in the models the errors show the mentioned dipole pattern. We extended the calculation of MAEs by adding a 200 m margin onto the observation-derived BLHs, which fully includes the dipole structures.
- o As a result, the MAEs of air temperature over sea ice are indeed closer together between ERA5 and CARRA (14% larger in ERA5). However, the MAE of specific humidity over open ocean has almost the same pattern (62% larger in ERA5 than CARRA; previously it was 70%).
- o We updated  all sections of our article with regards to this.

Section 3.3.2: The diabatic heating and moistening profiles are very interesting. How do the differences between observations and reanalyses fit together with the differences in surface fluxes? For example, at the later stages of the MCAO evolution, the diabatic heating profiles in ERA5 show that the warming extends too far up. This suggests that overall there is too much energy input into the BL, which in turn seems inconsistent with an underestimation of the surface sensible heat fluxes.

- o That is a very good observation. We think that the main reason lies again in the cloud scheme of ERA5 vs CARRA. In some of the literature we now cite (namely, Brümmer et al. 1996 and Tetzlaff et al. 2015), the relevance of entrainment fluxes is detailed. We had already investigated the diabatic cooling observed in some altitudes based on ERA5 temperature tendencies in the supplement, and suggested that the cooling is not a net radiative cloud top cooling, but instead a sign of turbulence. Of course, a turbulence-driven mixing of the ABL with warmer air masses aloft leads to a cooling of the upper air masses, while at the same time heating of lower air masses. Furthermore, increased condensation into clouds implicates also increased heating rates.

- We now discuss all of these issues, starting by directly mentioning the inconsistency which at first arises between SSHFs and heating rates.

L408ff: I can't fully reconcile the statement that ERA5 overestimates cloud liquid water content at all stages. For the time range 1-2h the values shown in Fig. 9b seem to agree quite well with observations, or am I missing something?
- That statement was indeed not ideal. We rephrased to
  *"ERA5 tends to over-estimates the amount of liquid water …"*.
- Furthermore, we put slightly more emphasis on the increased precipitation in ERA5, also adding the new radar data from Figure 9a into the discussion. The modeling of precipitation can be much more challenging than modeling of clouds. Thus, it is to no surprise for us that the climatological investigation showed systematic differences between ERA5 and CARRA precipitation.

Generally, I feel that some of the figures presented in the Appendix are interesting but not key to the understanding of the study (specifically figures A3 - A8). Hence, I suggest moving them from the Appendix to the Supplement along with the corresponding text.
- We agree. The figures were moved from the Appendix to a separate Supplement.

**Editorial comments**

l2: *intensive -> intense*
l8: delete *specific* in investigated *specific* MCAO
l11: Suggest to rephrase *As the air mass continued its drift southwards,* ... as *With increasing fetch of the sea ice edge, …*
l12: *quasi-Lagrange -> quasi-Lagrangian*
l16: ... issues with **the representation** of ...
l63: **at** high temporal resolution
l99: *as **a** dedicated*
l108: *several  such flight legs*
l110: *reported -> performed*?
l120: vertical gradient (?)
l125: Please fix the reference ECMWF, 2016 by including the appropriate authors
l154: *The reanalysis data can be retrieved for...*
l189: Please spell out LAGRANTO (Lagrangian Analysis Tool)
l236: replace *obvious* by *evident*
l241: *liquid+ice -> liquid and ice*
l245: replace *issue* by *matter*
l246: the sea ice concentrations **in** ERA5 **are**
l274: please rephrase *respective observations derived BLH*
l294: ... *vertical velocity is used **for** the three-dimensional...*
l404: here and elsewhere: *cloud ice **and** snow*
l405: fix ref to Maherndl et al (2023)
l423: *common intensity* – Do you mean *typical intensity*?
L453: *resolution* (singular)
Caption Fig. 2: ... *diamond shapes show the location**s** of released* ...
Caption Fig. 3: ... *indicated **on** the left hand side* ... (please also fix in other captions)
- Thank you for all above editorial comments; all suggested changes have been implemented